# ACTIVE TOPOLOGICAL MAPPING BY METRIC-FREE EXPLORATION VIA TASK AND MOTION IMITATION

## ABSTRACT

Topological map is an effective environment representation for visual navigation. It is a graph of image nodes and spatial neighborhood edges without metric information such as global or relative agent poses. However, currently such a map construction relies on either less-efficient random exploration or more demanding training involving metric information. To overcome these issues, we propose active topological mapping (ATM), consisting of an active visual exploration and a topological mapping by visual place recognition. Our main novelty is the simple and lightweight active exploration policy that works entirely in the image feature space involving no metric information. More specifically, ATM's metric-free exploration is based on task and motion planning (TAMP). The task planner is a recurrent neural network using the latest local image observation sequence to hallucinate a feature as the next-step best exploration goal. The motion planner then fuses the current and the hallucinated feature to generate an action taking the agent towards the hallucinated feature goal. The two planners are jointly trained via deeply-supervised imitation learning from expert exploration demonstrations. Extensive experiments in both exploration and navigation tasks on the photo-realistic Gibson and MP3D datasets validate ATM's effectiveness and generalizability.

## 1 INTRODUCTION

Mobile agents often create maps to represent their surrounding environments [6]. Typically, such a map is either topological or metrical (including hybrid ones). We consider a topological map to be metric-free, which means it does not explicitly store global/relative position/orientation information with measurable geometrical accuracy [39, 38]. Instead, it is a graph that stores local sensor observations, such as RGB images, as graph nodes and the spatial neighborhood structure (and often navigation actions) as graph edges that connects observations taken from nearby locations. While metric maps are often reconstructed by optimizing geometric constraints between landmarks and sensor poses from classic simultaneous localization and mapping (SLAM), topological maps have recently attracted attention in visual navigation tasks due to the simplicity, flexibility, scalability, and interpretability [58, 13, 27, 40, 12].

A topological map used for visual navigation could be constructed in two ways. The first and simplest way is to let the agent explore the new environment through metric-free *random walks*, after which the map could be built by projecting the recorded observations into a feature space and adding edges between nearby or sequentially obtained features [58]. However random walk is very inefficient especially in large or complex rooms, leading to repeated revisits of nearby locations in the same area. The other way is to design a navigation policy that controls the agent to more effectively explore the area while creating the map. It is known as *active SLAM* and often involves some metric information as either required input [42, 12] or intermediate estimations [13]. As shown in Fig. 1, could we combine the merits of the two ways by finding *a metric-free (neither input nor estimates) exploration policy that discovers informative traversing trajectories in unknown environments* for topological map construction after exploration?

To achieve this objective, we propose Active Topological Mapping (ATM) as shown in Fig. 2. It contains two stages: active exploration through a learned metric-free policy, and topological mapping through visual place recognition (VPR) [51]. The first stage adopts the task and motion planning formalism (TAMP) [26, 55] and imitation learning [63] from expert demonstrations which could come from either an *oracle policy* having full access to virtual environments, or simply a *human expert* in real world. Our main novelty is to design such an imitation at both the task and the motion levels with joint end-to-end training. Our task planner, a two-layer LSTM [31] network trained with deep supervision, conceives the next best goal feature to be

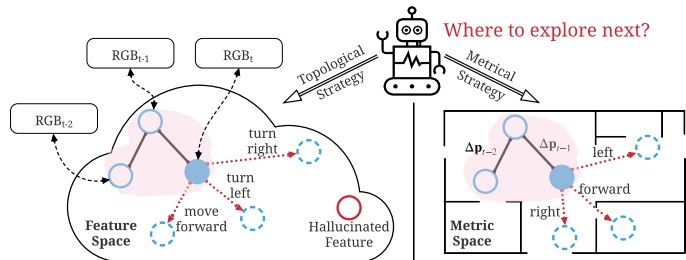

Figure 1: **Problem overview**. We focus on the active mapping problem where a mobile agent needs to decide how to efficiently explore a novel environment. For planning and navigation, we embrace the topological feature space where each feature corresponds to an image observation, while the metric space involves distance/pose information which is onerous to obtain accurately. Our main idea is to hallucinate goal features to guide exploration actions, learned by imitating expert demonstrations.

explored by hallucination from the current and historical image features. Our motion planner, a simple multi-layer perceptron, fuses the current and the hallucinated features and generates the best action that will move the agent to a location whose feature is closer to the goal.

The second stage of ATM takes all observations recorded during the active exploration stage to create the topological map. This stage could be solved similar to [58], where nodes are first connected by the sequential visiting order, and then additional node connections are discovered by a binary classifier estimating the spatial adjacency between two nodes through their image similarity. Differently, we adopt VPR, a classic technique in SLAM for loop closure detection, to discover additional edges more effectively. We further train an *action assigner* to assign each newly-added edge with corresponding actions that will move the agent between the two connected nodes. Finally, the topological map becomes our efficient environment representation for visual navigation as in [58].

We validate the efficacy of ATM on two tasks: *exploration* in which the goal is to maximize the explored area within a fixed step budget, and *navigation* in which the goal is to use ATM-constructed topological map to navigate the agent to a target image. In summary, our contributions are:

- We propose a simple and effective framework named as active topological mapping (ATM) for efficient and lightweight visual exploration. The topological map it constructs can be used for efficient visual navigation.

- We develop joint trainable feature-space task and motion planning (TAMP) networks to achieve metric-free and generalizable exploration.

- We design a deeply-supervised imitation learning strategy to train the feature-space TAMP networks with better data efficiency.

- We validate our method on the photo-realistic Gibson [72] and MP3D [9] datasets in both visual exploration and navigation.

## 2 RELATED WORK

**Topological map in exploration and navigation.** Previous works tried to tackle navigation with end-to-end learning of sensorimotor control by directly mapping visual observations to the action space [57]. However, such purely reactive RL-based methods that have no explicit memory struggle to navigate in complex scenarios [20, 74]. Newer methods that tackle this problem with scene memory [23, 29] often rely on metric information. An explicit metric map is commonly used for localization and navigation in the literature [22, 21, 6], but may face robustness and computation challenges, especially in dynamic and complex scenes, due to the need for accurate geometric constraints during the map and pose optimization. Later, inspired by the animal and human psychology [70], researchers show that topological map may aid robot navigation [13, 15, 11, 5, 29]. In literature, many topological mapping solutions either uses a random walkthrough sequence [58], or incrementally constructs a topological graph during the navigation task [13, 40]. However, random exploration is inefficient in creating a comprehensive map given a limited exploration time. And the existing exploring-while-mapping solutions still involve metric information either as required input or as intermediate estimation. Instead, we propose a two-stage solution that (1) learns an efficient and generalizable exploration policy to collect visual observations of a novel environment, and (2) uses VPR to construct a topological map for future navigation. Similar exploration-before-navigation pipelines include [5]

Figure 2: **Workflow of ATM.** ATM consists of two stages: (1) **active exploration** of a novel environment by task and motion planning in feature space; and (2) **topological mapping** composed of initialization via temporal adjacency and completion via visual place recognition (VPR).

and [54], yet they still need metric information to create the topological map and have lower data efficiency due to the use of RL.

**Exploration in navigation.** While various studies have focused on passive map-building and target-driven navigation policy learning, active SLAM has received less attention: how to actively control the camera for map construction [25]. Recent works [35, 14, 12] use learning-based methods to solve the problem of active exploration. [12] learns a hierarchical policy for active exploration via both classic and RL methods. [16] learns policies with spatial memory that are bootstrapped with imitation learning and finally finetuned with coverage rewards. [66] proposes a curiosity-based RL method which uses episodic memory to form the novelty bonus. Different from existing methods, we propose task and motion imitation which learns metric-free, generalizable, and data-efficient exploration.

**Hallucinating future latent feature.** The idea of hallucinating future latent features has been discussed in other application domains. Previous work has utilized this idea of visual anticipation in video prediction/human action prediction [71, 73, 10, 24, 68], and researchers have applied similar ideas to robot motion and path planning [33, 37, 8, 60]. As stated in [71, 73, 68], visual features in the latent space provides an efficient way to encode semantic/high-level information of scenes, allowing us to do planning in the latent space, which is considered more computationally efficient when dealing with high-dimensional data as input [46, 32]. Different from previous robotics work, we take advantage of this efficient representation by adding deep supervision when anticipating the next visual feature, which was computationally intractable if we were to operate at the pixel level.

**Deeply-supervised Learning** has been extensively explored [41, 67, 44, 45] during the past several years. The main idea is to add extra supervisions to various intermediate layers of a deep neural network in order to more effectively train deeper neural networks. In our work, we adopt similar idea to deeply supervise the training of feature hallucination and action generation

**Task and motion planning.** Task and motion planning (TAMP) divides a robotic planning problem into: the high-level task allocation (task planning) and the low-level action for the task execution (motion planning). This framework is adopted in many robotic tasks such as manipulation [17, 53] exploration [7] and navigation [49, 69]. Such framework allows us to leverage high-level information about the scenes to tackle challenges in local control techniques [4]. In this work, to perform active topological mapping of a novel environment, the agent firstly reasons at the highest level about the regions to navigate: hallucinate the next best feature point to visit. Afterward, the agent takes an action to get to the target feature. The whole procedure is totally implemented in feature space without any metric information.

## 3    ACTIVE EXPLORATION BY IMITATION IN FEATURE SPACE

Our topological map is represented by a graph $\mathcal{G} = (\mathcal{I}, \mathcal{A})$, where the graph nodes denoted by $\mathcal{I}$ is a set of RGB panoramic image observations collected by the agent at different locations $\mathcal{I} = \{I_1, I_2, \cdots, I_N\}$ (where $N$ denotes the number of nodes), and the edges denoted by $\mathcal{A}$ is composed of a set of actions $a_{(I_i, I_j)} \in \mathcal{A}$ which connects two spatially adjacent observations $I_i$ and $I_j$. In this work, each RGB panoramic image is of size $256 \times 512$, and the action space consists of three basic actions: move_forward, turn_left, and turn_right. Active exploration aims at maximizing the topological map coverage over an environment given a certain step budget $N$. The coverage of the topological map denoted by $\mathcal{C}$ is defined as the total area in the map that is known to be traversable or non-traversable. Mathematically, let $\pi_\theta$ denote the policy network parameterized by $\theta$, $a_t$ denote the action taken at step $t$, and $\Delta \mathcal{C}(a_t)$ denote the gain in coverage introduced by taking action $a_t$, the following objective function is optimized to obtain the optimal exploration policy $\pi^*$:

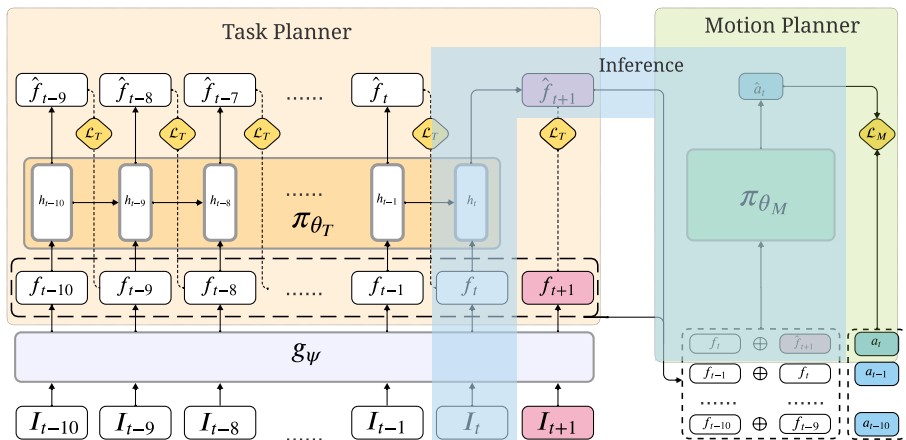

Figure 3: **Training and inference for task and motion imitation.** Feature extractor $g_\psi$ takes image $I_t$ as input and generates the corresponding feature vector $f_t$. *TaskPlanner* $\pi_{\theta_T}$ is a recurrent neural network (RNN) consuming a sequence of features $\{f_{t-10}, \cdots, f_t\}$ to hallucinate the next best feature to visit $\hat{f}_{t+1}$. *Motion-Planner* $\pi_{\theta_M}$ consumes the concatenation (denoted by $\bigoplus$) of $f_t$ and $\hat{f}_{t+1}$ and generates the action to move the agent towards the hallucinated feature. During training, we supervise all the intermediate outputs including the intermediate hallucinated features $\{\hat{f}_{t-9}, \cdots, \hat{f}_t\}$ and the intermediate actions $\{\hat{a}_{t-10}, \cdots, \hat{a}_{t-1}\}$, in addition to the final output $\hat{f}_{t+1}$ and $\hat{a}_t$. During inference, current observation $I_t$ is firstly encoded and fed into $\pi_{\theta_T}$ to hallucinate $\hat{f}_{t+1}$, and then $\hat{f}_{t+1}$ combined with the $f_t$ is fed into $\pi_{\theta_M}$ for motion planning. $\mathcal{L}_T$ is $L_2$ loss and $\mathcal{L}_M$ is cross entropy loss (the subscripts $T$ and $M$ denote **T**ask and **M**otion respectively). $h_t$ denotes the hidden state of RNN.

$$\pi^* = \arg\max_{\pi_\theta} \mathbb{E}_{a_t \sim \pi_\theta} \Big( \sum_{t=1}^{N} \Delta\mathcal{C}(a_t) \Big) . \tag{1}$$

**Learning from expert demonstrations.** In literature, most works solve Eq. (1) by reinforcement learning to maximize the reward [16, 43], such solutions are not only data-hungry, but also require complicated training involving metric information. Differently, we adopt imitation learning [63] to let our policy network $\pi_\theta$ mimic the output of the expert policy $\tilde{\pi}$ which could come from either an *oracle policy* having full access to virtual environments, or simply a *human expert* in real world. Hence, our objective is to minimize the difference between our policy network and the expert policy:

$$\pi^* = \arg\min_{\pi_\theta} \mathcal{L}(\pi_\theta, \tilde{\pi}) , \tag{2}$$

where $\mathcal{L}$ measures the discrepancy between two policies. We propose a novel task and motion imitation in feature space to solve Eq. (2) which will be introduced in the following sections. We respectively introduce the feature extraction (3.1), the policy network $\pi_\theta$ composed of a *TaskPlanner* denoted by $\pi_{\theta_T}$ (3.2) as well as a *MotionPlanner* denoted by $\pi_{\theta_M}$ (3.3), and the deeply-supervised learning strategy (3.4).

### 3.1 FEATURE EXTRACTION

We firstly encode each visual observation $I_t \in \mathcal{I}(t = 1, 2, ..., N)$ with a feature extractor $g_\psi$ parameterized by $\psi$ which uses the ImageNet [64] pre-trained ResNet18 backbone [30]. The feature embedding $f_t \in \mathbb{R}^d(d = 512)$ is obtained by $f_t = g_\psi(I_t)$, (see Fig. 3). Note that $g_\psi$ is jointly optimized with the task planner $\pi_{\theta_T}$ as well as the *MotionPlanner* $\pi_{\theta_M}$ via imitation learning.

### 3.2 TASK PLANNER FOR NEXT BEST FEATURE HALLUCINATION

*TaskPlanner* $\pi_{\theta_T}$ parameterized by $\theta_T$ takes the most recent $m$-step visual features $\mathcal{F} = \{f_{t-m}, \cdots, f_t\}$ as input, and learns to hallucinate the next best feature to visit which is denoted by $\hat{f}_{t+1}$, see Fig. 3. In specific, $\pi_{\theta_T}$ is a two-layer LSTM [31]:

$$\hat{f}_{t+1} = \pi_{\theta_T}(f_{t-m}, \cdots, f_t | \theta_T) . \tag{3}$$

To save computation, $\pi_{\theta_T}$ only takes the most recent $m$-step features as input and we empirically find that $m = 10$ achieves good performance. In other words, *TaskPlanner* is only equipped with a short-term scene memory, and it tries to extend the feature space as quickly as possible in order to guide the agent to perform efficient exploration.

### 3.3 MOTION PLANNER FOR ACTION GENERATION

*MotionPlanner* $\pi_{\theta_M}$ parameterized by $\theta_M$ takes the hallucinated feature $\hat{f}_{t+1}$ and the current feature $f_t$ as input, and outputs the action taking the agent towards the hallucinated goal (see Fig. 3). Specifically, $\pi_{\theta_M}$ is a multi-layer-perceptron (MLP) taking the concatenation of two features as input to classify the action:

$$\hat{a}_t = \pi_{\theta_M}(\hat{f}_{t+1}, f_t | \theta_M) . \tag{4}$$

### 3.4 DEEPLY-SUPERVISED IMITATION LEARNING STRATEGY

Our imitation pipeline is shown in Fig. 3. Given an expert exploration demonstration including a sequence of images and the corresponding expert actions $\mathcal{E} = \{\{I_1, a_1\}, \{I_2, a_2\}, \cdots, \{I_N, a_N\}\}$, we adopt the deeply-supervised learning strategy [41] to jointly optimize the feature extractor $g_\psi$, task planner $\pi_{\theta_T}$, and *MotionPlanner* $\pi_{\theta_M}$. Ultimately, our objective in Eq. (2) becomes,

$$\min_{\psi, \theta_T, \theta_M} \sum_{t=1}^{N-1} \mathcal{L}_T(\hat{f}_{t+1}, f_{t+1}) + \sum_{t=1}^{N} \mathcal{L}_M(\hat{a}_t, a_t) , \tag{5}$$

where $\mathcal{L}_T$ is $L_2$ loss to measure the discrepancy between two features, and $\mathcal{L}_M$ is cross entropy loss to make the model imitate the expert action category. The desired target feature $f_{t+1}$ is obtained by $f_{t+1} = g_\psi(I_{t+1})$ ($I_{t+1}$ is obtained from the expert demonstration $\mathcal{E}$), the desired action $a_t$ is also read from $\mathcal{E}$, the hallucinated feature $\hat{f}_{t+1}$ is calculated by Eq. (3), and the generated action $\hat{a}_t$ is computed by Eq. (4). For each training iteration, we randomly clip $m + 1$ observations and the corresponding $m$ actions from an expert exploration ($m = 10$ and $N \gg m$), and feed them to $g_\psi$, $\pi_{\theta_T}$, and $\pi_{\theta_M}$. Note that we supervise all the intermediate output, as illustrated in Fig. 3.

**Remark.** Different from other works, we decompose the exploration policy $\pi_\theta$ into a task planner $\pi_{\theta_T}$ as well as a *MotionPlanner* $\pi_{\theta_M}$, and jointly learn them by imitation in feature space. *TaskPlanner* hallucinates high-level features and *MotionPlanner* generates low-level actions. We supervise all the intermediate output of the network including task-level features $\hat{f}_t$, thus our TAMP imitation strategy can be considered as deeply-supervised learning [41].

## 4 TOPOLOGICAL MAPPING VIA VISUAL PLACE RECOGNITION

The topological map $\mathcal{G} = (\mathcal{I}, \mathcal{A})$ initialized by the active exploration experience in Sec. 3 is unidirectionally connected in the temporal domain. Each node (a panoramic RGB image observation) is just connected with its preceding node and next node, failing to discover spatial adjacency. We propose to further complete the initial map $\mathcal{G}$ by adding edges to the node pairs if two nodes in a pair possess high visual similarity. Specifically, we adopt visual place recognition (VPR) [34, 3] method and geometric verification to measure "visual similarity" of a node pair.

Given a room of interest and all the nodes' images, for every node, we first used non-learning, VLAD-based VPR system to pick out other images that are very similar to this node as candidates for further geometric verification. After this VPR process, we conduct geometric verification by inspecting if the number of matching SIFT keypoints between the node's image and a candidate's image is larger than a threshold (in our case 100). Passing the threshold means that we will add edges between these two nodes. Meanwhile, we train a model named *ActionAssigner* to assign an action list to each new edge. The architecture of *ActionAssigner* is similar to *MotionPlanner*, yet *ActionAssigner* predicts a sequence of actions with two node features as input, while *MotionPlanner* is a one-step action predictor (predict just one action).

We adopted VPR based method mainly due to its time efficiency compared to methods involving brute force, such as that in SPTM [58] or brute-force geometric verification. See Appendix F for more details on the VPR process we used, its time efficiency, and our implementation of SPTM.

Our completed topological map can perform image-goal navigation tasks. Given the image observations for the start and goal positions, we localize them on topological map by retrieving the node image with the highest inlier number (via VPR). Once localized, we apply Dijkstra's algorithm [19] to find the shortest path between the two nodes. We can then smoothly navigate the agent from the start position to the goal position without metric information.

## 5 EXPERIMENTS

We test ATM on two datasets: Gibson [72] and Matterport3D (MP3D) [9] dataset on Habitat-lab platform [52]. The two datasets are collected in real indoor spaces by 3D scanning and reconstruction methods. The agents can be equipped with multi-modality sensors to perform various robotic tasks. The average room size of MP3D ($100\ m^2$) is much larger than that of Gibson ($[14m^2, 85m^2]$).

We run experiments on two tasks: (1) **autonomous exploration** proposed by Chen *et al.* [16], in which the target is to maximize an environment coverage within a fixed step budget (1000-step budget following [12]), and (2) **image-goal navigation** where the agent uses the constructed topological map to navigate from current observation to target observation. Regarding the exploration, we employ two evaluation metrics: (1) coverage ratio which is the percentage of covered area over all navigable area, and (2) absolute covered area ($m^2$). We exactly follow the setting by ANS [12] that a point is covered by the agent if it lies within the agent's field-of-view and is less then $3.2m$ away. Regarding the navigation, we adopt two evaluation metrics: shortest path length (SPL) and success rate (Succ. Rate) [2]. We again follow ANS [12] to train ATM on Gibson training dataset (72 scenes), and test ATM on Gibson validation dataset (14 scenes) and MP3D test dataset (18 scenes). Using the ATM trained on Gibson dataset to test on MP3D dataset helps to show ATM's generalizability.

### 5.1 EXPERIMENT CONFIGURATION

**Exploration setup.** In exploration, we independently explore each scene 71 times, each time assigning the agent a random starting point (created by a random seed number). We keep track of all the random seed numbers for result reproduction. We use the Habitat-lab sliding function so that the agent does not stop when it collides with a wall, but instead slides along it. In order to generate the initial 10 steps required by ATM, we constantly let the agent to execute `move_forward` action. Once it collides the wall, it randomly chooses `turn_left` or `turn_right` action to continue to explore. Afterwards, we iteratively call *TaskPlanner* and *MotionPlanner* to actively explore the environment. During ATM-guided exploration, we allow the agent to actively detect its distance with surrounding obstacles or walls. When the agent's forward-looking distance to obstacles is less than 2-step distances and the ATM predicted action is still `move_forward`, the agent will turn to a new direction (by randomly iteratively executing `turn_left` and `turn_right` action) with the longer forward-looking distance to new obstacles.

We experiment with two locomotion setups: the first one is with step-size 0.25 m and turning angle $10°$, which follows the same setting established in [12] for comparing with baseline methods in the exploration task. The second one is with step-size 0.30 m and turn-angle $30°$. This helps us test ATM's generalization capability under different locomotion configurations.

**Navigation setup.** In navigation, we encourage the agent to visit enough positions for each room scene. Specifically, the agent has collected 2,000 images per room on Gibson and 5,000 images per room on MP3D (2,000/5,000-step ATM-guided exploration).

**Expert demonstrations.** To generate expert demonstrations, we place so-called anchor points relatively evenly over all navigable areas in each training room. The agent starts at a random anchor point and walks to the closest anchor point with help from Habitat's API. We repeat this process until all the anchor points are visited. During this process, we record all the actions the agent takes, along with all the panoramic photos at each time step. Note that such a demonstration is not a strong demonstration, as this process is not directly related to the task of maximizing exploration or efficient navigation. For more details on expert demonstration, please refer to Appendix D and Fig. 5

**Training details.** In our implementation, the local observation sequence length is 10 (m=10) because we find it has achieved good performance-memory trade-off. We experimentally tested $m = 20$ and got inferior performance. ATM network architecture is in Appendix E (parameter size is just 16 M). We train ATM with PyTorch [61]. The optimizer is Adam [36] with an initial learning rate 0.0005, but decays every 40 epochs with the decaying rate 0.5. In total, we train 70 epochs. We train all the ATM variants with the same hyperparameter setting for fair comparison.

## 5.2 COMPARISON METHODS

For exploration task, we compare ATM with six RL-based methods: 1. **RL + 3LConv**: An RL Policy with 3 layer convolutional network [52]; 2. **RL + Res18**: RL Policy initialized with ResNet18 [30] and followed by GRU [18]; 3. **RL + Res18 + AuxDepth**: adapted from [56] which uses depth map prediction as an auxiliary task. The network architecture is the same as ANS [43] with one extra deconvolution layer for depth prediction; 4. **RL + Res18 + ProjDepth** adapted from Chen *et al.* [16] who project the depth image in an egocentric top-down in addition to the RGB image as input to the RL policy. 5. **ANS** (Active Neural SLAM [12]) jointly learns a local and global policy network to guide the agent to explore; 6. **OccAnt** [65]: takes RGB, depth and camera as input to learn 2D top-down occupancy map to help exploration. For ablation studies, we have following ATM variants:

1. **RandomWalk** The agent randomly choose an action to execute at each step. It serves as a baseline and helps us to know agent exploration capability without any active learning process. Please note that RandomWalk is also the SPTM [58] exploration strategy.

2. **ATM_noFeatHallu.** We use the architecture of *TaskPlanner* to directly predict the next action. It discards task planning in feature space, but instead plans directly in action space. Its performance helps us to understand if the hallucinated feature is truly necessary.

3. **ATM_withHistory.** ATM is trained with only a short-memory (the latest $m$ steps observations). To validate the influence of long-term memory, we train a new ATM variant by adding extra historical information: we evenly extract 10 observations among all historically explored observations excluding the latest $m$ steps. After feeding them to ResNet18 [30] to get their embedding, we simply use average pooling to get one 512-dimensional vector and feed it to *TaskPlanner* LSTM as the hidden state input.

4. **ATM_LSTMActRegu.** *TaskPlanner* hallucinates a next-best feature at each time step to deeply supervise (or regularize) the whole framework in feature space. As an alternative, we can predict action instead at each time step (regularize action). This ATM variant helps us to figure out whether supervising each step of *TaskPlanner* LSTM in feature space is helpful.

5. **ATM_noDeepSup.** ATM without deeply-supervised learning. We remove LSTM per-frame feature supervision in *TaskPlanner* and neighboring frame action supervision in *MotionPlanner*. In other words, we just keep the feature prediction and action classification between the latest step and future step. It helps to test the necessity of involving deeply-supervised learning strategy.

## 5.3 EVALUATION RESULTS ON EXPLORATION

The quantitative results on exploration task are shown in Table 1. We can observe from this table that ATM achieves comparable performance on both Gibson dataset with RL-based methods, and best-performing result on MP3D dataset by outperforming all RL-based methods significantly (about $13\%$ coverage ratio and $40m^2$ area improvement). Since the comparing RL-based methods [1, 56, 16, 12] build the map in metric space and requires millions of training images, ATM is desirable because (1) it provides a metric-free option for exploration, and (2) it is lightweight (in terms of parameter size 16 M) and requires much less training data (just about 0.45 million images, in contrast with 10 million images required by most RL-based methods). The room scenes in MP3D dataset are much more complex and larger than those in Gibson dataset. They contain various physical impediments (e.g. complex layout, furniture), and some rooms contain outdoor scenarios. Hence, **ATM exhibits stronger generalizability in exploring novel scenes than RL-based methods**. Moreover, performance gain is more obvious on both Gibson and MP3D datasets when we change the agent to a different locomotion setup (from 0.25/10° to 0.30/30°), which also shows ATM is robust to different locomotion setups.

On Gibson dataset, ATM achieves lower coverage ratio than ANS [12] but higher average covered area. We find such performance difference is mainly caused by ATM stronger capability in exploring large areas than RL-based methods. In most cases, ATM actively reaches to new area within limited steps. This result is also echoed by the coverage ratio curve between ATM and ANS [12] (Fig. III in Appendix C), from which we can see ATM quickly increases the coverage ratio during the early exploration stage (higher curve slope). If we use a different locomotion setup (0.30/30°), ATM coverage ratio stays as highest along the exploration process (left-most sub-figure of Fig. III).

**Comparison with random exploration.** *RandomWalk* serves as the baseline for our framework. It is also adopted by SPTM [58] to build topological map. It involves no learning procedure and the agent randomly takes an action at each step to explore an environment. From Table 1, we can see that *RandomWalk*

Table 1: **Quantitative results of exploration task** over 1000-step budget. Top three performances are highlighted by red, green, and blue color, respectively.

| Method Description | Method | Sensor Used | #Train Imgs | Gibson Val | | Domain Generalization MP3D Test | |
|---|---|---|---|---|---|---|---|
| | | | | %Cov. | Cov. $(m^2)$ | %Cov. | Cov. $(m^2)$ |
| Non-learning Based | RandomWalk (used by SPTM [58]) | No | No | 0.501 | 22.268 | 0.301 | 40.121 |
| RL-based | RL + 3LConv [1] | RGB, Depth, Pose | 10 M | 0.737 | 22.838 | 0.332 | 47.758 |
| | RL+ResNet18 | | 10 M | 0.747 | 23.188 | 0.341 | 49.175 |
| | RL+ResNet18+AuxDepth [56] | | 10 M | 0.779 | 24.467 | 0.356 | 51.959 |
| | RL+ResNet18+ProjDepth [16] | | 10 M | 0.789 | 24.863 | 0.378 | 54.775 |
| | OccAnt [65] | | 1.5-2 M | **0.935** | **31.712** | 0.500 | 71.121 |
| | ANS [12] | | 10 M | **0.948** | **32.701** | 0.521 | 73.281 |
| ATM Model Variants | ATM_NoDeepSup | RGB | 0.45 M | 0.768 | 26.671 | 0.292 | 37.163 |
| | ATM_LSTMActRegu | | | 0.914 | 35.238 | 0.610 | 101.734 |
| | ATM_withHistory | | | 0.917 | 35.331 | **0.618** | **102.302** |
| | ATM_NoFeatHallu | | | 0.907 | 34.563 | 0.589 | 99.091 |
| Deeply Supervised Imitation | **ATM (0.25 m/10°)** | | | 0.918 | 35.274 | **0.642** | **109.057** |
| | **ATM (0.30 m/30°)** | | | **0.927** | **37.731** | **0.656** | **117.993** |

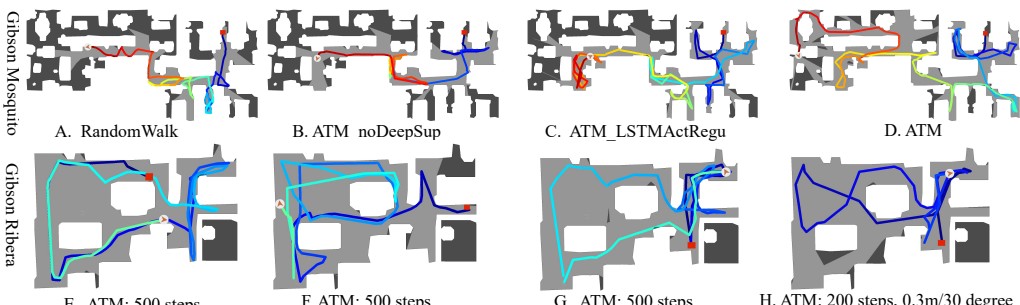

Figure 4: **Exploration trajectories Visualization**. Top row: various ATM variants exploration result (1000-step budget) on Gibson `Mosquito` scene. Bottom row: exploration with different start positions (E, F, G, 500-step budget, with agent step-size 0.25 m and turn-angle 10°). Agent with larger step-size and turn-angle (0.3 m/30°) achieves similar coverage ratio with much smaller steps (200 steps, F). The trajectory color evolving from cold (blue) to warm (yellow) indicates the exploration chronological order. More exploration results are given in Appendix A, Fig. I

dramatically reduces the exploration performance in terms of both coverage ratio and average coverage area. The inferior performance of *RandomWalk* verifies the necessity of learning active exploration strategy in order to help agent to efficiently explore an environment. Figure 4 demonstrates the qualitative comparison between *RandomWalk* and ATM exploration result.

**Feature regularization and with history memory**. If we replace feature regularization involved in *TaskPlanner* with action regularization (*ATM_LSTMActRegu*), we have observed more performance drop on MP3D than on Gibson dataset (3% versus 0.2%), which shows adopting feature regularization improves the generalizability compared with action regularization. Moreover, introducing full history memory (*ATM_FullHistory*) to *TaskPlanner* (used as LSTM hidden state input) produces very similar results on Gibson dataset, but significantly reduces the performance on MP3D dataset (more than 2% drop). It thus shows using historical memory tends to encourage ATM to overfit training data, so that its generalizability is inevitably reduced. We argue that such generalizability drop might lie in our over-simplified history memory modelling because we just evenly sample 10 nodes (image observations) from all historically visited nodes, which might be too simple to represent the whole history memory, or even confuses *TaskPlanner* if the agent has already explored many steps. More elegant long-term history memory model remains to be explored.

**Deeply-supervised learning and joint task and motion imitation.** Removing deeply-supervised learning (*ATM_noDeepSup*) leads to obvious performance drop on both Gibson and MP3D dataset. In MP3D datast, it evens leads to worse performance than *RandomWalk*. It thus shows the necessity of deep supervision in both feature space (*TaskPlanner*) and action space (*MotionPlanner*). Meanwhile, *ATM_noFeatHallu* leads to significant performance drop on both Gibson and MP3D datasets. It thus attests the advantage of our feature-space task and motion imitation strategy which jointly optimize *TaskPlanner* for high-level task allocation and *MotionPlanner* for low-level motion control.

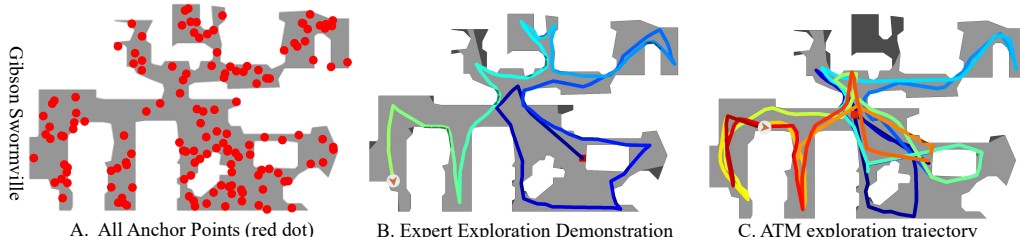

A. All Anchor Points (red dot)   B. Expert Exploration Demonstration   C. ATM exploration trajectory

Figure 5: **Expert Demonstration and ATM exploration visualization**. **A**: all potential anchor points in Gibson Swormville scene. **B**: expert selectively traverses a subset of the anchor points (choose one anchor to traverse among a group of close anchors) to cover the whole area. **C**: ATM exploration trajectory with 1000-step budget, with the model learned from expert demonstration.

Table 2: **Quantitative results of navigation tasks**. Top three performances are in **red**, **green**, and **blue** color, respectively. We collect 2000 (Gibson)/5000(MP3D) images with agent setup (0.25m/10°).

| Method | Gibson Val | | Domain Generalization on MP3D Testset | |
|---|---|---|---|---|
| | Succ. Rate (↑) | SPL (↑) | Succ. Rate (↑) | SPL (↑) |
| RandomWalk | 0.027 | 0.021 | 0.010 | 0.010 |
| RL + Blind | 0.625 | 0.421 | 0.136 | 0.087 |
| RL + 3LConv + GRU [1] | 0.550 | 0.406 | 0.102 | 0.080 |
| RL + ResNet18 + GRU | 0.561 | 0.422 | 0.160 | 0.125 |
| RL + ResNet18 + GRU + AuxDepth [56] | 0.640 | 0.461 | 0.189 | 0.143 |
| RL + ResNet18 + GRU + ProjDepth [16] | 0.614 | 0.436 | 0.134 | 0.111 |
| IL + ResNet18 + GRU | 0.823 | 0.725 | **0.365** | **0.318** |
| CMP [28] | **0.827** | **0.730** | 0.320 | 0.270 |
| OccAnt (RGB) [65] | 0.882 | 0.712 | N/A | N/A |
| ANS [12] | **0.951** | **0.848** | **0.593** | **0.496** |
| ATM | **0.957** | **0.859** | **0.733** | **0.619** |

We also visualize the comparison between ATM hallucinated next-step future feature and truly observed feature in Appendix Fig. II (C). We see that the hallucinated feature is more similar with the observed real feature when the agent is walking through a spacious area (in other words, the agent mostly takes move_forward action), than when the agent is walking along room corner, against the wall or through narrow pathway. This may be due to the learned *TaskPlanner* most likely hallucinates feature moving the agent forward if the temporary egocentric environment allows. This also matches expert exploration experience because experts (like humans) mostly prefer moving forward so as to explore as more area as possible.

## 5.4 EVALUATION RESULTS ON NAVIGATION

The navigation result is in Table 2. We compare ATM with most of methods compared in exploration task. CMP [28] builds up a topdown belief map for joint planning and mapping. For OccAnt [65], we just report its result with model trained with RGB image (so as to be directly comparable with ATM). We can see that ATM outperforms all comparing methods on the two datasets, with the largest performance gain on MP3D dataset (about 14% Succ. Rate, 12% SPL improvement). Hence, we can conclude that our built topological map can be used effectively for image-goal navigation. More importantly, ATM exhibits satisfactory generalizability in navigation owing to both active exploration in feature space and topological mapping in VPR. In Fig. II (B), we can see VPR and *ActionAssigner* successfully add new edges (purple lines) to spatial-adjacent RGB images (nodes), resulting in a spatial-adjacency representative topological map. More detailed discussion on navigation is given in Appendix B.

## 6 CONCLUSION

We propose a novel active topological mapping (ATM) framework for visual navigation, which is lightweight and computationally efficient. Our ATM's active exploration is metric-free, and can be learned via deeply-supervised task and motion imitation learning with superior data efficiency and generalizability compared to other RL-based methods. We also provide some **real-world experiments to show our sim2real transfer-ablity** (in Appendix H). Future works include designing elaborate historic memory modules and involving multi-modality sensors to further improve the performance of visual exploration and navigation.

## 7 REPRODUCIBILITY STATEMENT

Please refer to Section. 5.1. on how to reproduce our expert demonstration dateset. Appendix. E. provides details on how to implement all the neural networks used in our work, and Appendix. F. provides details on how to implement the VPR system used in the topological map construction. All the code used in this paper are shared through Supplementary Material.

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

APPENDIX

## A    MORE RESULTS ON EXPLORATION

We show an extra exploration result on MP3D [9] q9vSo1VnCiC room scene in Fig. I. In this figure, we show both the room scene 3D visualization in top row and multiple exploration results in the two bottom rows. Specifically, we show three efficient exploration results in the middle row, in which the agent begins to explore at different positions (marked by small red rectangle patch). We further show two less-effective exploration results in the bottom row (last two sub-figures), which are mainly caused by repetitive visiting within a local area (yellow-to-red trajectory).

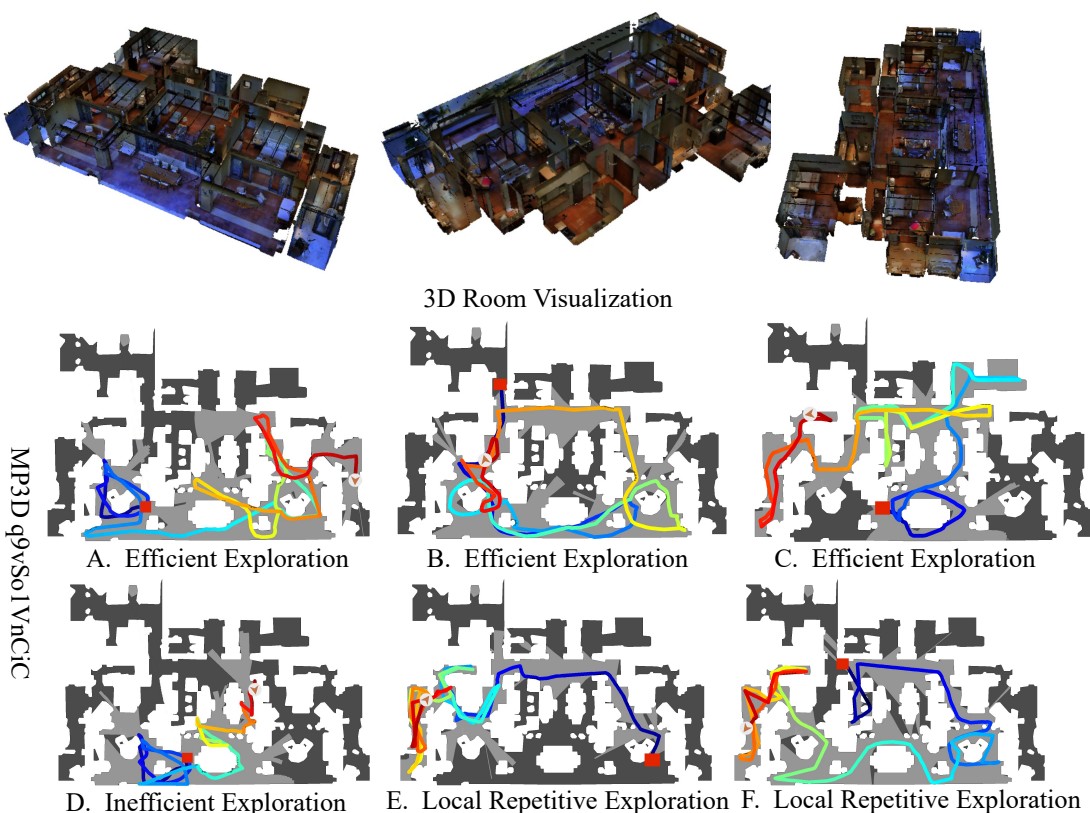

Figure I: ATM exploration result on MP3D [9] room scene q9vSo1VnCiC. We show both efficient exploration result (middle row, sub-figure A, B, C). We also show relatively less-efficient exploration in sub-figure E and F, which are mainly due to local repetitive exploration. We further show an inefficient exploration example in sub-figure D. The room scene 3D visualization is given in the top row. The agent exploration starting-position is marked by a red rectangle patch.

## B    MORE RESULTS ON NAVIGATION

**Visualizations of topological mapping.** We show a qualitative example of ATM in Fig. II. We can see from exploration trajectory that agent actively traverses all areas across multiple rooms in the environment scene, VPR and *ActionAssigner* sucessfully add new edges (purple lines in the middle figure) to spatial-adjacent nodes, resulting in a complete and robust topological map.

## C    COVERAGE RATIO CURVE

The coverage ratio curve over 1000-step budget between ATM (two locomotion settings) and ANS [12] is shown in Fig. III, from which we can see ATM quickly increases the coverage ratio during the early exploration stage.

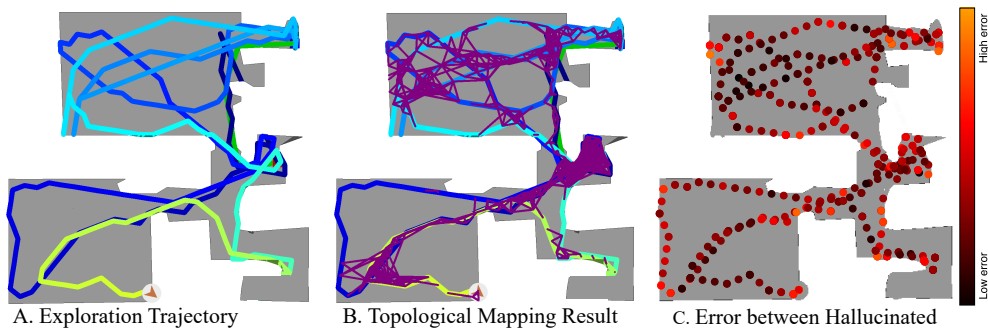

| A. Exploration Trajectory | B. Topological Mapping Result | C. Error between Hallucinated Feat. and Real-Observed Feat. |

Figure II: **Feature Visualization A**. The exploration trajectory (blue to yellow, step size 0.25 m and turn-angle $30°$) with 500-step budget, overlaid on top of floor plan map. **B**. The spatially-adjacent panoramic images are connected (purple lines) via VPR. **C**. The difference (Euclidean distance in 512-d feature space) between real-observed features and hallucinated features. The darker of the color, the lower of the difference.

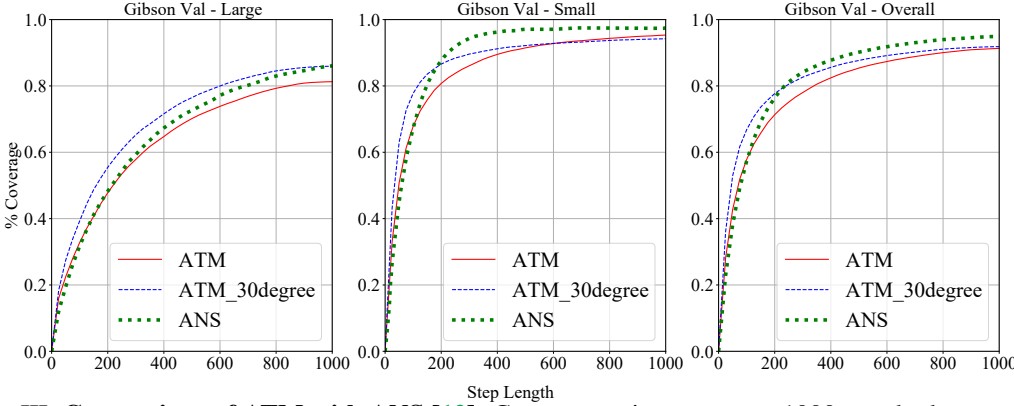

Figure III: **Comparison of ATM with ANS [12].** Coverage ratio curve over 1000-step budget over large $> 50m^2$, small $< 50m^2$ and all (average) room size, respectively.

## D  MORE DETAILS ON EXPERT DEMONSTRATION

We sample a set of anchor points across all navigable area in the each room scene, then call Habitat-lab `Pathfinder` API to let the agent traverse all anchor points to cover the whole area. `Pathfinder` API generates the most economic action sequence (following shortest-path geodesic path) leading the agent from one anchor point to another. In practice, we use the provided episodes' start/end points in Gibson as the anchor points since they densely cover all the navigable area (see Fig. 5). We collected 450 k training images and the associated actions in 72 scenes from Gibson, which are used to train both ATM and *ActionAssigner*.

## E  NEURAL NETWORK ARCHITECTURE

### E.1  ATM NEURAL NETWORK ARCHITECTURE

Our ATM topological mapping neural network architecture is given in Table I, it consists of ResNet18 (as the image embedding module), LSTM layer (for task planner) and FC layer (for task planner and motion planner).

### E.2  ACTIONASSIGNER NETWORK ARCHITECTURE

The *ActionAssigner* neural network is given in Table II. The *ActionAssigner* also uses ResNet18 as the image embedding layer. Then it uses a sequence of FC layers to predict a sequence of actions separately (sequence length is 6). Bidirectional LSTM is applied to model actions mutual dependencies. The parameter size 13.5 M. We train *ActionAssigner* with the same parameter setting as the famrwork of *TaskPlanner* and *MotionPlanner* (the network in Table I).

Table I: ATM network architecture illustration. The network consists of basic 2D image convolution layers, such as ResNet18, LSTM and FC. The network is lightweight, the parameter size is just 16 M.

| Layer Name | Filter Num | Output Size |
|---|---|---|
| **Image Embedding Layer** | | |
| Input: [10, 3, 256, 512] | | |
| Embedding Network: ResNet18 | | |
| Embedding Size: [10, 512] | | |
| **Task Planner Network** | | |
| LSTM | layers = 2, hiddensize = 512 | [10, 512] |
| Feat Prediction FC | in feat = 512, out feat = 512 | [10, 512] |
| **Motion Planner Network** | | |
| Input: Feat [10, 1024], Action: [10] | | |
| Feat Merge FC | in feat = 1024, out feat = 512 | [10, 512] |
| Action Classification FC | in feat = 512, out feat = 3 | [10, 3] |

Table II: *ActionAssigner* neural network architecture. Given two panoramic image observations, we predict six sequential actions in total.

| Layer Name | Filter Num | Output Size |
|---|---|---|
| **Image Embedding Layer** | | |
| Input: [2, 3, 256, 512] | | |
| Embedding Network: ResNet18 | | |
| **Feat. Merge Layer** | | |
| Concat. Size: [1, 1024] | | |
| FC | in feat = 1024, out feat = 512 | [1, 512] |
| **Action Predict Branch** | | |
| branch1 FC | in feat = 512, out feat = 128 | [1, 128] |
| branch2 FC | in feat = 512, out feat = 128 | [1, 128] |
| branch3 FC | in feat = 512, out feat = 128 | [1, 128] |
| branch4 FC | in feat = 512, out feat = 128 | [1, 128] |
| branch5 FC | in feat = 512, out feat = 128 | [1, 128] |
| branch6 FC | in feat = 512, out feat = 128 | [1, 128] |
| **Action Predict** | | |
| Concat. Size: [1, 6, 128] | | |
| BiLSTM | layers = 1, out feat = 128 | [1, 6, 128] |
| Action Classify FC | in feat = 128, out feat = 3 | [1, 6, 3] |

## F    MORE DETAILS ON VISUAL PLACE RECOGNITION

### F.1    DETAILS ON THE IMPLEMENTATION OF VLAD-BASED VPR AND ACTION ASSIGNER

For VPR, we first extract SIFT[50] features from all the images captured during the exploration episode, and then compute VLAD feature for each image based on the clustered SIFT as described in [34]. This process involves running K-Means [48] on all the SIFT features aggregated. We used Scikit-Learn [62]'s K-Means implementation with `n_clusters=16, n_init=1`. We then store all the VLAD features into a ball tree [59, 47] with `leaf_size=60` and $L_2$ as distance metric. Finally, for every node, we use its image's VLAD feature to query the ball tree for 20 nearest neighbors as candidates. All the hyperparameters remains unchanged during all the testing, and we have not done any hyper-parameter searching.

We show the VPR result in Fig. IV where C. is the candidates picked by the VPR process and we can see VPR can successfully reflects image observations' spatial closeness. In practice, we again use Habitat-lab `pathfinder` to get the ground truth actions between any two VPR-connected image observations. If the obtained action list length is smaller than six, we append STOP action in the end. If the obtained action list length is larger than six, we remove the connection. In navigation task, the agent just stops once it sees the predicted STOP action from the inferred action sequence by the *ActionAssigner*.

### F.2    TIME EFFICIENCY OF VPR

In Table. III, we demonstrate the time efficiency superiority of our VPR system. The "Time Spent" indicates the amount of time each method takes to finish retrieving all node pairs that should be connected (to add edge) in a room. Specifically, we report the average time cost across all the 14 test rooms in the Gibson dataset (excluding *ActionAssigner* time cost). The resulting maps after spatial-adjacency connection created by the three methods are almost identical. We can see that our proposed VPR-based method has the minimal time cost (or time complexity).

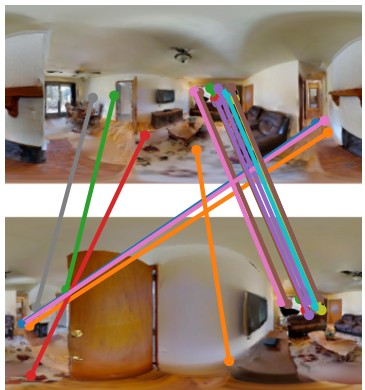 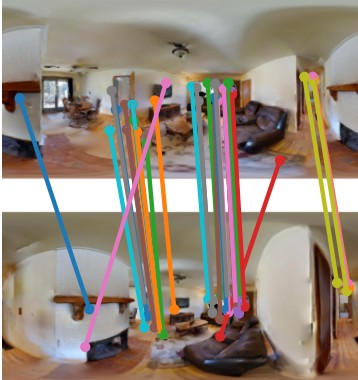 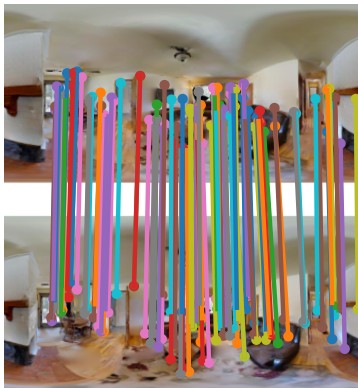

A.  Non-similar Image Pair (15 steps)   B.  Semi-similar Image Pair (5 steps)   C.  Similar Image Pair (2 steps)

Figure IV: VPR result visualization on Gibson Cantwell environment scene. The colored lines indicate paired inlier points between two image. We use the the same reference figure (top row), and compute its VPR result with three images the agent takes at another location (different step location). Left: 15 steps difference with nearly dissimilar image pair, middle: 5 steps difference with semi-similar image pair, right: 2 steps difference with the most similar image pair.

| Methods | Time Spent |
|---|---|
| SPTM [58] | $\approx 2.1$ hr |
| Brute-Force Geometric Verification | $\approx 26$ hr |
| **VLAD-Based VPR (Ours)** | $\approx 0.2$ hr |

Table III: The average time spent on constructing a topological map given all the images of a room

In this experiment, We implemented the retrieval network in the SPTM [58] to do classification between every two image on whether they should be connected. The network architecture and hyper-parameters are the same as in the paper, with some small tweaks to the architecture's dimension due to the fact the original paper is done in a low-resolution video game while our testing environment is a photo-realistic dataset. The memory graph in SPTM [58] is similar to the concept of topological graph in our setting, and SPTM locomotion network is similar to our `ActionAssigner`.

The Brute-Force Geometric Verification method exhaustively apply geometric verification on all image pairs. Its time cost can be potentially reduced by turning to parallelism computation.

All the experiments are done with 10 cores of a Intel Xeon Platinum 8268 (205W, 2.9GHz), 32GB RAM and a SSD. SPTM [58] additionally run on a Nvidia V100 GPU with a batch size of 160.

## G   EXPLORATION EFFICIENCY DISCUSSION

An ideal exploration should constantly keep reaching to new area, like the expert exploration does. We find our proposed ATM framework sometimes leads to inefficient exploration: repetitively visiting a local area. We show three examples of such inefficient exploration on Gibson Swormville environment scene in Fig. V. Our introduced random action perturbation strategy can reduce the "repetitive visiting" dilemma to some extent. According to our understanding, such "repetitive visiting" is mainly caused by ATM "short-memory" when predicting the next action to take, and involvement of no global environment scene information, including the whole historically visited memory (scene memory) and topdown map (telling the agent if the area to explore has already been visited), to supervised the whole exploration process. We do not involve such extra supervision because we want to keep our framework as simple and lightweight as possible. It remains as a future research direction to explore how to involve extra information to achieve more efficient exploration. We also provide several video (in which left image shows the panoramic image observation and the right image shows the corresponding exploration trajectory update on the top-down map.)

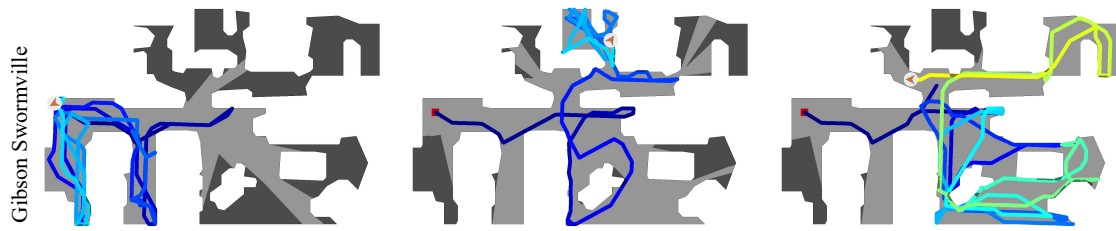

A. Trajectory on 500-step budget    B. Trajectory on 500-step budget    C. Trajectory on 700-step budget

Figure V: Examples of inefficient exploration visualization on Gibson Swormville environment scene (with agent 0.3m/30° setup). We can see that ATM inevitably leads to repetitive visiting of local area. We argue that such such "repetitive visiting" is mainly caused by the "short memory" (just using the most recently visited short image observation sequence to predict the next action) of ATM, and exploitation of no global scene information, such as the assist of full history memory and global swiftly-updated top-down map.

## H    MORE DETAILS ON REAL-WORLD HARDWARE EXPERIMENT

### H.1    HARDWARE DETAILS

See Fig. VI. for the physical setup of the robot we built. We added an aluminium frame on top of a iRobot Create 2 to host a Nvidia Jetson TX2 as the processor and an Insta 360 Pro2 to capture panoramic images. The turning angle ($10 \deg$), step length(0.25m), camera height(1.5m) and other important parameters are designed to be as close to the configuration in the simulation as possible. An example image captured by the camera is given in VIII.

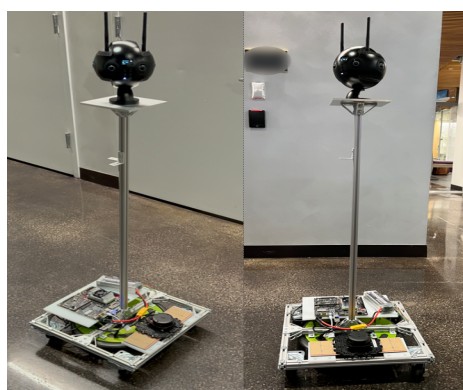

Figure VI: Two different angles of the robot

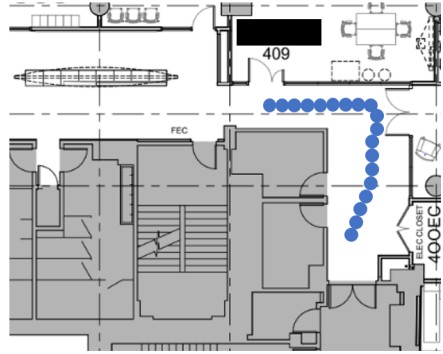

Figure VII: The hand-drawn trajectory of the 50-step exploration episode.

### H.2    REAL-WORLD EXPERIMENT DETAILS

With the help of our ATM model (with no real-world fine-tuning), we conducted a preliminary experiment using this robot to explore an indoor environment. The result is in VII The step budget is capped at 50 steps (vs. 1000 in simulation) due to limitations in battery capacity and prolonged execution time caused by the fact that the camera does real-time onboard images stitching to produce panoramic images. We are continuously improving the hardware system by finding better way to capture panoramic images and enhancing the stability of the camera by adding more support structures, and hope to deliver more hardware experiment results.

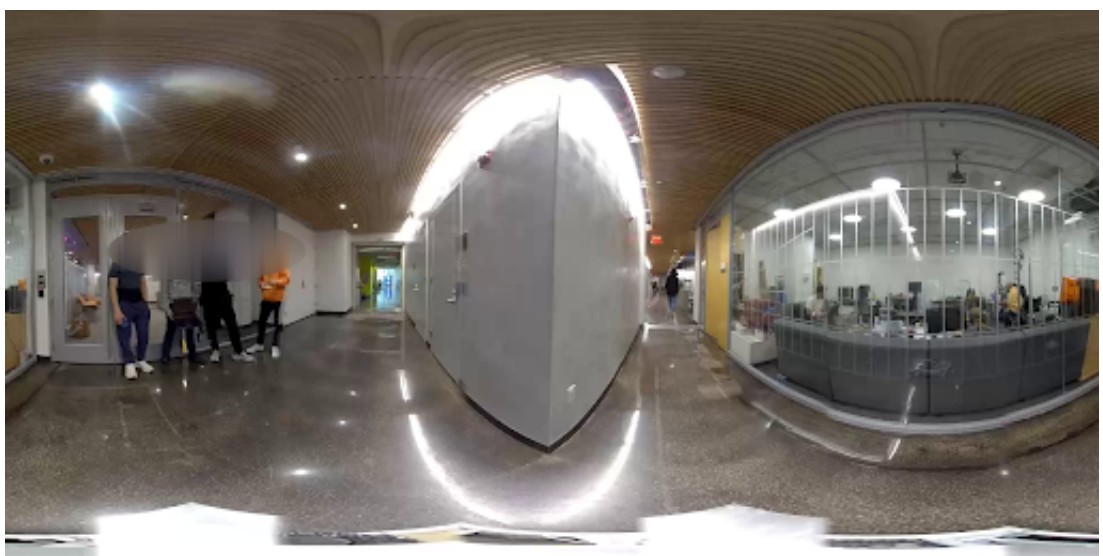

Figure VIII: One of the image captured by our Insta 360 Pro2 camera while in exploration

