# OpenReview forum: "Active Topological Mapping by Metric-Free Exploration via Task and Motion Imitation"
_ICLR.cc/2023/Conference — Submitted to ICLR 2023_

### Official Review · Reviewer_ZEmd · 2022-10-23

**Confidence:** 3
**Correctness:** 3
**Technical Novelty And Significance:** 2
**Empirical Novelty And Significance:** 2
**Recommendation:** 3

**Clarity, Quality, Novelty And Reproducibility:**

Although the proposed framework in this work is self-motivated and presented clearly, there are some missing details and confusing parts:

(1) In the navigation experiment, the number of images collected during the exploration stage would affect the navigation results significantly. Why is the agent allowed 2k/5k steps to collect 2k images per room on Gibson and 5k images per room on MP3D? Is it based on the coverage of the exploration area? Did authors conduct ablation studies on this hyperparameter?

(2) Regarding the ATM_noDeppSup baseline, after removing the feat regularization loss in the task planner and action regularization loss in the motion planner, how to train the model? Based on Equ (5), these two losses are the only training signal to train the model. Does the policy degenerate into a random policy?

(3) Since MP3D has a larger area in the scene, why not train the policy on the MP3D environment, then evaluate on the Gibson environment?

(4) What does the ActionAssigner perform on the validation or testing set? Can authors share some evolution numbers on it? Since the constructed topological map is metric-free, this ActionAssigner should play an essential role in the downstream task (e.g., navigation).

(5) In Figure II (C), the authors show the error between hallucinated and real-observed features. Can authors share some quantitative evaluation on the hallucinated feature? It is unclear how to interpret what a policy learns in a very high-dimension latent space. Can authors share some insights regarding it?

**Strength And Weaknesses:**

+) The idea of adopting TAMP to the exploration policy design is self-motivated. Moreover, the studied problem, efficient exploration over an unseen scene, is also a meaningful direction. Finally, the final experiments also demonstrate the effectiveness of the proposed framework.

+) The paper is well-written and easy to follow. The teaser figure shows the studied problem clearly, and the following framework figure and model figure illustrate the implantation in detail. Several qualitative results and videos in the main paper and supplementary present a clear visualization.

o) In the “exploretraj_MP3D_q9vSo1VnCiC_1.mp4” video, could authors illustrate why the agent keeps circling at a local area for a while? Is the agent keep executing move_forward and colliding with a wall?

-) The main idea of designing an efficient exploration policy is based on hallucinating features at the next step. However, this idea is similar to visual forecasting or anticipation, and there is no discussion regarding it in the related work. Many works focus on using visual input to predict the feature at the next step in a latent space, such as [1, 2, 3, 4, 5]. There are also many works utilizing future predictions for manipulation or navigation.

-) Although the experimental results demonstrate that the overall framework performs well on the exploration and navigation tasks, the model design and the way to hallucinate features at the next step are well developed. Therefore, the technical contributions in this work are incremental. Nevertheless, it would be appreciated if the authors could explore or share more insights about how to utilize the hallucinate features for motion planning.

[1] Carl Vondrick et al., “Anticipating Visual Representations from Unlabeled Video”, CVPR 2016.

[2] Kuo-Hao Zeng et al., “Visual Forecasting by Imitating Dynamics in Natural Sequences”, ICCV 2017.

[3] Chien-Yi Chang et al., “Procedure Planning in Instructional Videos”, ECCV 2020.

[4] Basura Fernando et al., “Anticipating human actions by correlating past with the future with Jaccard similarity measures”, CVPR 2021.

[5] Dídac Surís et al., “Learning the Predictability of the Future”, CVPR 2021.

**Summary Of The Paper:**

This work follows task and motion planning (TAMP) to design a high-level task planner and a low-level motion planner for more efficient exploration of an indoor scene. The goal is to cover the scene as efficiently as possible to construct a topological map. After the exploration, an off-the-shelf visual place recognition (VPR) system is used to find more edges on the initially constructed map. Finally, the trained action assigner predicts the possible action list causing the transition on the newly detected edges. The main contributions in this work are three-fold:

(1) Design an efficient exploration policy based on TAMP.

(2) Predict a possible next hidden state (imitate expert trajectories) at the task planner and use it as input to the motion planner.

(3) Further, leverage an off-the-shelf VPR method and a learned action assigner to establish the completed topological map reconstruction system.

The experiments on the exploration and the navigation tasks in the Gibson and MP3D environments demonstrate the effectiveness of the proposed framework.

**Summary Of The Review:**

This work proposes to learn an efficient exploration policy by way of TAMP. The main idea is to train a task planner to hallucinate the next state in a feature space so that the motion planner can predict the action accordingly. Although the idea is self-motivated, the studied problem is essential, and the paper is well-presented with good results, the technical contributions and overall novelty are incremental. In addition, there is a missing discussion on visual forecasting and its application to planning and control in the related work section. Therefore, I would suggest the authors revise the paper based on them.

---

> ### Author Response · Authors · 2022-11-19
> **Response to  Reviewer ZEmd**
>
> Thank you sincerely for the great feedback! We provide our detailed responses as follows:
>
> **Q1: The idea of hallucination is similar to visual forecasting or anticipation, and there is no discussion regarding it in the related work. Therefore, the technical contributions in this work are incremental. Nevertheless, it would be appreciated if the authors could explore or share more insights about how to utilize the hallucinate features for motion planning.**
>
>
> A1: Thank you for pointing this out. We have added a discussion on those methods in our updated paper (labelled in blue color).
> We did not discuss those in the first submission because of (1) the space limit of the paper, and (2) we found those to be a bit far away from this work due to the fact that they are used in different application domains. However, after careful reading of the reference you generously provided, we believe they are very informative and pave the way for our explanation of our novel contribution of adding deep supervision. As many of your proposed references pointed out, latent features encode semantic information of a scene in an efficient format without many irrelevant pixel values. We further take advantage of this efficient representation to add deep supervision to our Task Planner, which does planning in the latent space. We can potentially do deep supervision in the pixel space, but that would require much more computation, and a good chunk of such computation will be responsible for figuring out high-level information out of pixel values. **Thus, using the idea of feature hallucinations to enable deep supervision is one of the keys to our data efficiency. As you can see in the review of 9Cpk, our technique is not widely known in the robotics field, even for experts, and we think this contribution deserves to be recognized by the ICLR community.**
>
> **Q2: Explain the circling in exploretraj_MP3D_q9vSo1VnCiC_1.mp4**
>
> A2: The agent keeps circling around a local area because our framework ATM just uses the most up-to-date local visual observation sequence to infer the next action, without access to the global map or full historical memory. So it sometimes leads to the agent repetitively visiting a local area, especially for some difficult regions, which is indeed not optimal. We are working on improving this further in our future work after this paper, but we believe the submitted results are reasonably good as Table 1 shows because our average performance is still comparable to or even better than baseline methods that use metric maps.
>
>
> **Q3: Why is the agent allowed 2k/5k steps to collect 2k images per room on Gibson and 5k images per room on MP3D? Is it based on the coverage of the exploration area? Did authors conduct ablation studies on this hyperparameter?**
>
> A3: In exploration, we followed the established experiment protocol (the step budget 1000) for a fair comparison with the baseline methods.
> In navigation, since there are not many explore-then-navigate methods (therefore no established protocol), we allow the agent to travel more steps in order to cover more areas. Navigation on the pre-built topological map requires the topological map to cover as much area as possible. We find 2k/5k step exploration gives a good covered area for the navigation tasks. Note that the SPTM method needs to travel >>2K steps for their random exploration, which is inefficient compared to us.
>
>
> **Q4: ATM_noDeppSup baseline, after removing the feat regularization loss in the task planner and action regularization loss in the motion planner, how to train the model? Does the policy degenerate into a random policy?**
>
>
> A4: In ATM_noDeepSup, we just remove the regularization loss between step $t-10$ and step $t$ in the task planner. The loss for the step $t$ to $t+1$ still exists (the loss for the next-step feature hallucination). The whole framework is still end-to-end trainable and does not fall back to a degenerated random policy, as shown in the ATM_noDeepSup row in Table 1, which has better performance than the random policy. We have rewritten this section in Section 5.2 (text shown in blue color).
>
>
> **Q5: Why not train the policy on the larger MP3D environment, then evaluate on the Gibson environment?**
>
>
> A5: Reason 1: We have to follow the established experiment protocol as in ANS; Reason 2: Gibson environment is simpler than MP3D. Training on the Gibson environment and then testing on the MP3D environment can better show the strong generalization capability of a method. (simpler-to-harder task transfer)

---

> > ### Author Response · Authors · 2022-11-19
> > **Response to Reviewer ZEmd**
> >
> > **Q6: What does the ActionAssigner perform on the validation or testing set? Can authors share some evolution numbers on it? Since the constructed topological map is metric-free, this ActionAssigner should play an essential role in the downstream task (e.g., navigation).**
> >
> >
> > A6: The ActionAssigner is responsible for assigning the edge between a newly connected two nodes (temporally unconnected but newly connected by VPR ) with the corresponding action list that leads the agent from the first node to the second node. The ActionAssigner is trained on the training dataset used for training ATM. More discussion is given in Sec. D.2 of the Appendix (page 16).  A more holistic evaluation of this module’s performance is implicitly included in our navigation evaluations.
> >
> >
> > **Q7: Can authors share some quantitative evaluation on the hallucinated feature? It is unclear how to interpret what a policy learns in a very high-dimension latent space. Can authors share some insights regarding it?**
> >
> >
> > A7: Thank you for this question. We are also very interested in figuring out why or how planning in a feature space leads to better performance. We tried to visualize it in Figure II in the appendix (page 16 top), and it seems like the hallucinated features are slightly more accurate when the robot is farther away from obstacles like walls. But to be honest, just like many existing deep-learning-based methods (including the compared baseline methods), trying to interpret the learned network is a non-trivial task on its own. So currently, we don’t have any decisive answer or further insight. We would welcome any suggestion that would help us interpret the learned policy better in an understandable way.

---

### Official Review · Reviewer_9Cpk · 2022-10-24

**Confidence:** 5
**Correctness:** 3
**Technical Novelty And Significance:** 2
**Empirical Novelty And Significance:** 2
**Recommendation:** 3

**Clarity, Quality, Novelty And Reproducibility:**

The novelty and originality of the work are a concern since the proposed framework is generally based on a pipeline of existing techniques. On the other hand, the paper is well written, and each component in the pipeline is well described.

**Strength And Weaknesses:**

Strength:

+ Topological mapping is an important problem for agent navigation.
+ The paper is well written; each method in the pipeline is well described.

Weakness:

- Novelty and contribution to address active mapping is a key concern.
-- The paper seems to argue the contribution is on "active" mapping. However, it is not clear how exactly the approach allows an agent to "actively" map the environment. The agent simply uses the policy learns from a demonstrator, and no active actions are taken during the execution to map the environment.
-- Each component also uses or marginally extends existing methods, and the novelty of each component is not well justified. For example, the idea of using a network to learn representations of the environment and use the representations to generate agent actions is commonly used; the VPR component also uses the previous methods.

- The term Task Planner in Section 3.1 is very confusing. The approach seems to simply predict the future feature vector in the next time step given a short sequence of the past features. What is the definition of tasks? Or what is the difference between task planning and feature prediction?

- Although the related work section reviews task and motion planning mainly for manipulation, the paper does not review and compare with a large body of local planning and control methods for navigation in the robotics community, which is more relevant to the paper’s problem domain.

- Why predicting the future feature vector in the next step is necessary? If the approach is able to predict a high-dimensional feature vector in the next step, then why the approach cannot directly predict the low-dimensional future action in the next step?


**Summary Of The Paper:**

This paper presents a framework of topological mapping, which is implemented as a pipeline of two main components including a planner to generate robot navigational actions and a visual place recognition (VPR) method to construct a topological map.

**Summary Of The Review:**

The paper proposes to address an important problem of active mapping for agent navigation. However, novelty of the paper is a main concern: it is not clear why the topological mapping is "active" and the novelty of each component used in the pipeline is also low.

---

> ### Author Response · Authors · 2022-11-19
> **Response to Reviewer 9Cpk**
>
> Thank you sincerely for the great feedback! We provide our detailed responses as follows:
>
> **Q1: it is not clear how exactly the approach allows an agent to "actively" map the environment. The agent simply uses the policy learned from a demonstrator, and no active actions are taken during the execution to map the environment.**
>
>
> A1: In active SLAM and active perception in general, the term “active” means the algorithm is not passively receiving sensor input for map estimation or other perception tasks, but to “actively” plan/control the sensor movement (https://arxiv.org/pdf/2207.00254.pdf; https://ieeexplore.ieee.org/document/5968).
> In our work, the term “active” is clear because our policy takes past sensing observations and outputs an action that controls the robot/sensor movement, and the goal of this policy is to explore unknown environments as quickly and efficiently as possible while without any metric input or metric quantity estimations.
> *To answer your questions directly, there is indeed active action taken during the execution of the policy: based on the current image and the hidden state, the policy tells the robot what “action” to take in the next step in order to move the camera to explore the environment.*
>
>
> **Q2: the difference between task planning and feature prediction (The term Task Planner in Section 3.1 is very confusing. The approach seems to simply predict the future feature vector in the next time step, given a short sequence of the past features. What is the definition of tasks? Or what is the difference between task planning and feature prediction?)**
>
>
> A2: The task and motion planning formulation is taken from the Task and Motion Planning (TAMP) framework that is widely used in the robotics area. It divides a robot planning problem into two main parts: high-level task allocation (task planning) and low-level action for the task execution (motion planning), as we discussed in the related work section.
>
> For the topological mapping problem in our work, high-level task planning means *figuring out which region of the feature space should be explored*, while low-level motion planning means *finding which immediate action to take in order to bring the robot closer to that region*.  Feature prediction via ConvLSTM is the way we implement this task planning (of course, there could be other implementations, which goes beyond the scope & claims of this paper).
>
> This is not a trivial implementation, as agreed by other reviewers such as bznr (“novelty could be the additional loss term to regularise the output of the motion planner”), because we connect topological mapping and feature-space planning: existing Active SLAM methods have to maintain a metric map in order to perform such a task planning (which is a metric goal-point prediction step in those methods), but such a metric map is not available in topological mapping, so we propose the feature prediction step to achieve active topological mapping.
>
> We appreciate this question and have added the above discussion in the paper to clarify the definition of “task” for future readers.
>
> **Q3: Lack of review for local planning and control methods for navigation in the robotics community**
>
> A3: Thanks for pointing it out. We added the relevant literature review in our updated version of the Related Work section in Task and motion planning. We added two related works [1], [2]. However, we would greatly appreciate it if you could generously provide us with a few related works as a lead to this field of literature.
>
> [1] Somil Bansal, Varun Tolani, Saurabh Gupta, Jitendra Malik, and Claire Tomlin. Combining optimal control and learning for visual navigation in novel environments. 2019.
>
> [2] Chao Cao, Hongbiao Zhu, Howie Choset, and Ji Zhang. TARE: A Hierarchical Framework for Efficiently Exploring Complex 3D Environments. In Proceedings of Robotics: Science and Systems, Virtual, July 2021.

---

> > ### Author Response · Authors · 2022-11-19
> > **Response to Reviewer 9Cpk**
> >
> > **Q4: Why predict the future feature vector? If the approach is able to predict a high-dimensional feature vector in the next step, then why don’t it directly predict the low-dimensional future action in the next step?**
> >
> >
> > A4: Thanks for this great question! It actually relates to the novel finding of this paper, and also it is the reason why this paper should NOT be rejected.
> >
> > **The short answer**: In the experiment result in Table 1, we directly show that predicting in the feature space together with action prediction leads to superior performance than directly predicting only the low-dimensional future action (ATM_noFeatHallu).
> >
> > **The long answer**: Predicting in feature/latent space and performing deep supervision of not only the moving action at the end of the network but also the intermediate latent features allows us to train an efficient exploration policy for topological mapping, and this is the key technical novelty in this paper. Without this technique, one can only supervise the final moving action as mentioned in your question, which is shown to be less effective in Table 1.
> > There are two key ideas:
> >
> > 1. As reviewer ZEmd pointed out, predicting a visual feature and then do decision making based on such hallucinated features is an idea that has been adopted in the domain of video prediction, human action and etc. Essentially, previous works have shown that latent features encode semantic information of a scene in an efficient format (reference can be found in the Related Work section). The Task Planner that learns from the features generated by the ground-truth sequence of the robot walking is then planning in the latent space, and this space has no noisy pixels that are irrelevant to the task at hand. The prediction gives out can then be more informative to the downstream motion planner. Again, this phenomenon is not our discovery. It’s been discussed in the works suggested by reviewer ZEmd. We have added them in the newly added “Hallucinating future latent feature” section in Related Works. This brings us to our second point.
> >
> > 2. This efficient representation brought by planning in latent space allows us to afford deep supervision. As you can see in the ablation study, we can technically only update at the end of the network, but we believe it would be less efficient in updating network weights of the earlier layers, due to well-known issues like vanishing gradients. While such issues might be alleviated by choosing a shallower network architecture with fewer layers, a shallower network would not have the necessary expressiveness for solving this topological exploration problem.
> > Therefore, there is a dilemma: a shallower network is not enough to solve the problem, while a deep network cannot reach its full potential if trained with only the action classification loss in the end.
> > The solution to this dilemma is our method. And we believe this question well justifies the contribution of our work and shows that our findings are non-trivial even to experts in this field.

---

### Official Review · Reviewer_CpyF · 2022-10-25

**Confidence:** 3
**Correctness:** 3
**Technical Novelty And Significance:** 3
**Empirical Novelty And Significance:** 2
**Recommendation:** 5

**Clarity, Quality, Novelty And Reproducibility:**

The work is original and the writing is clear. While the authors included many details of their approach in their work, I did not see that they will release the code.

**Strength And Weaknesses:**

## Strengths
* The approach outperforms all baselines on the exploration and navigation tasks. The authors choose reasonable baselines for the exploration task.
* The authors provide many details about their approach and architectures in the main paper and appendix.

## Weaknesses
* It seems strange that the locomotion setup has big impact on performance. Is this the case for the baseline approaches as well, or is the proposed approach particularly sensitive to this?
* Missing baselines for navigation (Sec. 5.4). The experiments in this section can be more thorough. For example, [1], [2], [3], [4], [5] are all good baselines to compare against, as they build a topological memory from exploration trajectories and use them for navigation (e.g. image-goal).
    * In appendix E.2 I find it surprising that SPTM takes so long to run. Is the topological map very dense?

## Notes
* Some additional related works on topological map construction and navigation are [1] and [2].
    * [1] is also similar to Savinov et al. They construct a topological map and connect edges using reachability. They utilize an exploration trajectory and then build a topological map from that.
    * [2] In particular, the topological map here is similar to the proposed approach where each node is a panorama. They utilize an exploration trajectory and then build a topological map from that.

* [1] Meng et al. 2020. Scaling local control to large-scale topological navigation.
* [2] Chen et al. 2021. Topological Planning with Transformers for Vision-and-Language Navigation.
* [3] Chaplot et al. 2020. Neural topological SLAM.
* [4] Savinov et al. 2018. Semi-parametric topological memory for navigation.

**Summary Of The Paper:**

The authors propose an approach for visual exploration using RGB cameras only. The approach is also able to build a topological map of the environment. To do the exploration, the authors use imitation learning to hypothesize the image feature in the next step and choose an action based on that feature (and current feature). For topological mapping, the authors use visual place recognition (VLAD and SIFT), which they claim is very efficient.

**Summary Of The Review:**

The proposed approach outperforms the baselines for the exploration task, but I think the navigation task experiments could be more thorough, especially since the topological map is one of the main contributions of the paper and this is how the constructed topological map is used.

---

> ### Author Response · Authors · 2022-11-19
> **Response to Reviewer CpyF**
>
> Thank you sincerely for the great feedback! We provide our detailed responses as follows:
>
> **Q1: strange that the locomotion setup has a big impact on performance. Need to clarify if this impact the baseline approaches as well.**
>
>
> A1: The two locomotion setups enable the agent to have different mobility setups: the first one is baseline setup (the same as the compared methods): stepsize 0.25 m and turn-angle $10^\circ$. The second setup gives the agent higher mobility: stepsize 0.30 m and turn-angle $30^\circ$. So it is naturally expected the agent will cover more area with the higher-mobility setup than the baseline setup, under the same step budget. We added the second setup to show the generalization capability of ATM under various locomotion settings. *Please note that even if we remove the second setup, our major experimental findings are not changed: ATM still achieves comparable performance on Gibson Val and better generalization performance on MP3D.*
>
>
> **Q2: Missing baselines for navigation**
>
> A2: Thanks for pointing this out. The main focus of our work is to propose a metric-free exploration framework that can actively construct a topological map, which could be used for many tasks, including visual navigation. The navigation task is just one way to show the usefulness of the built topological map.
>
> We can definitely compare with all the methods you proposed, but they are surely not as good because they do not involve an exploration phase that involves maximizing coverage ratio, so reporting those results together with that of the exploration-then-navigate methods will make the comparison unfair. If you insist on this, we could try our best, but please understand that our limited computing resources might not allow us to run all other navigation baseline experiments before the rebuttal deadline (definitely not before the paper revision deadline this Friday). Note that for [4] SPTM, we did include its random walk exploration in Table 1, and we include a brief time efficiency comparison in terms of connecting visually similar nodes in Appendix F.
>
>
>
> **Q3: I did not see that they will release the code.**
> We have submitted our code with the supplementary material, and we will surely open source the code via GitHub if this paper is accepted

---

### Official Review · Reviewer_bznr · 2022-10-25

**Confidence:** 4
**Correctness:** 3
**Technical Novelty And Significance:** 3
**Empirical Novelty And Significance:** 2
**Recommendation:** 6

**Clarity, Quality, Novelty And Reproducibility:**

The paper is generally well written. As mentioned in the previous section, however, I would like authors to present more clearly the baselines they introduce. Originality of the work is limited, but the simple contribution can be considered as solid if properly evaluated.

**Strength And Weaknesses:**

Strengths:
1. The core idea to hallucinate next goals directly in feature space is very interesting and promising.
2. The method is compared against many baselines and beats previous state-of-the-art approaches, both in exploration and ImageGoal navigation from the topological map built from data collected by the exploration policy.


Weaknesses:
1. A potential weakness of this work is the lack of technical originality in the proposed method. The topological map, mentioned in the paper, is similar compared with what has already been done in previous work. I’m not sure authors should put emphasis on the topological mapping part, both in the title, the abstract and the introduction, as there is actually no contribution on this side. Even in the experiment part, only the short Section 5.4 is about using the topical map. It is actually not clear whether the gain on the navigation task is about the use of a topological map, or simply the fact that the underlying exploration policy covers more of the scene. The architecture of the exploration policy is also standard, as well as the imitation learning objective.  The only novelty could be the additional loss term to regularise the output of the motion planner. However, if the gain from this loss term is clear (see next remark), this could be a great contribution.
2. Authors should make the description of the considered baselines in Section 5.2 clearer. In particular, is there a baseline that corresponds to the exact same architecture as the one proposed, but with only the action loss ($\mathcal{L}_M$) and no regularisation term (no $\mathcal{L}_T$)? This ablation is very important to showcase the importance of the contribution.
3. Authors mention that the necessity to have access to metrics information as done in previous work is a constraint, which can be true. However, their method necessitates access to strong expert data, which can be ok as current simulators can allow this, but they cannot claim that this is a lighter constraint.
4. In Section 5.1, authors mention "During ATM-guided exploration, we allow the agent to actively detect its distance with surrounding obstacles and walls". How is this implemented, and doesn't it make the comparison unfair with other SOTA methods ?

**Summary Of The Paper:**

This paper presents a method to explore an environment and collect data to then build a topological map of the considered scene. Such exploration policy is a simple lightweight recurrent baseline trained with imitation learning to mimic expert trajectories. The policy is composed of a feature extractor that encodes the observation, a task planner which is a 2-layer LSTM, and finally a motion planner that is an MLP outputting the action to take from the output of the motion planner. The main novelty in this work comes from an additional regularisation loss term that forces the output of the recurrent module (motion planner) to be as close as possible to the feature extracted from the next observation, i.e. enforcing the motion planner to output a next goal in feature space.

The exploration policy is compared with different baselines on the task of maximising floor coverage, and shows state-of-the-art performance on the challenging MP3D set of scenes. The data collected by the exploration policy is also used to build a topological map to perform the ImageGoal Navigation task.

**Summary Of The Review:**

The core idea introduced in this paper, i.e. hallucinating goals directly in feature space, is very interesting and promising. The lack of other novel ideas could be a weakness of the paper. However, if the gain brought by this additional regularisation is clearly showcased empirically, it can be a relevant contribution. As already mentioned, I want to be sure the ablation about keeping the exact full exploration policy architecture along with the imitation learning action loss term (exactly as it is used in the proposed method) and removing the regularisation term is properly done. I also have a few other concerns (stated in the "Weaknesses" section). I thus tend to consider the work as marginally below the acceptance threshold but am looking forward to clarifications from the authors.

---

> ### Author Response · Authors · 2022-11-19
> **Response to Reviewer bznr**
>
> Thank you sincerely for the great feedback! We provide our detailed responses as follows:
>
> **Q1: A potential weakness of this work is the lack of technical originality in the proposed method. The topological map mentioned in the paper is similar to what has already been done in previous work. I’m not sure the authors should put emphasis on the topological mapping part, both in the title, the abstract, and the introduction, as there is actually no contribution on this side. Even in the experiment part, only the short Section 5.4 is about using the topical map. It is actually not clear whether the gain on the navigation task is about the use of a topological map, or simply the fact that the underlying exploration policy covers more of the scene. The architecture of the exploration policy is also standard, as well as the imitation learning objective. The only novelty could be the additional loss term to regularise the output of the motion planner. However, if the gain from this loss term is clear (see next remark), this could be a great contribution.**
>
> A1: Thanks for pointing it out. We think there might be a **misunderstanding between topological maps and active topological mapping**. We clarify it here.
>
> We did not claim that our originality lies in the idea of using a topological map, it has been widely used already and could date back to several decades ago in the robotics community. **We are excited to claim novelty in how to explore an unknown environment to build such a topological map efficiently and effectively from only images without any metric inputs/estimates.**
>
> Despite the wide use of topological maps, this process of exploring and building topological maps, which we call active topological mapping, has been **underexplored** by all previous relevant works, which build the topological map either by **random walking** or by building a **metric map** first. *Our technical challenge (and novelty) is how to achieve this process in a pure feature space without relying on metric mapping.* Once achieved, this new process could bring some potential benefits, such as requiring less expensive 3D sensors or less demanding computations for metric mapping.
>
>
> **Q2: Not clear navigation gain is the use of a topological map.**
>
> A2: You are absolutely correct about “the gain on the navigation task” coming from “the fact that the underlying exploration policy covers more of the scene”, and this is exactly why we show the navigation performance comparisons: *it shows that our topological mapping process builds better topological maps than similar existing methods, achieving comparable performance than those using metric maps.* It also shows a **better generalization** of visual navigation using the topological map built by ATM. Theoretically, the topological map built by just random walking may still achieve good navigation performance if the agent is allowed to walk more (or unlimited) steps. We show ATM can build such a topological map with fewer steps (efficient exploration).
>
>
> **Q3: The architecture of exploration policy is also standard.**
>
> A3: We did not claim novelty in the architecture. What we claimed is **a deeply-supervised imitation learning strategy to train the feature-space TAMP networks with better data efficiency**. Previous learning methods in robotics do not adopt the deeply-supervised training but only supervise the network at the very end of the network. Although it might be effective for metric-based methods, such a learning strategy is shown in our work to be **inefficient in training feature-space TAMP networks for visual exploration**. *Thus, the architecture is standard, but the way to train it is new in our paper.*
>
>
> **Q4: Needs one more ablation study with the same architecture as the one proposed, but with only the action loss (L_m) and no regularisation term (L_t)?**
>
> We are following your advice to run the experiments; we will report the result once it comes out.

---

> > ### Author Response · Authors · 2022-11-19
> > **Response to Reviewer bznr**
> >
> > **Q5: Needs to demonstrate why expert demonstration is a lesser constraint than metric data**
> >
> > A5: In the real world, expert demonstration for topological exploration could be relatively easily achieved by *any human operator controlling the robot to explore some unknown environments*, without the need of creating metric maps of the environment, which could require well-trained SLAM or surveying experts. **This is because humans can explore and navigate in space without relying on accurate metric inputs.**
> >
> > It’s worth noting that our demonstration is not a strong expert demonstration as it’s not directly providing a demonstration of efficient exploration or navigation. It’s a process that can be relatively easily automated. We have modified our writing in Sec 5.1 to reflect this.
> > **Also, by only requiring RGB images as input, it is thus less constrained than methods requiring 3D sensors, which are usually more expensive than monocular cameras.** All these make topological-only methods appealing for real-world visual exploration/navigation, which we are working on now.
> >
> > Last, we actually never claimed or intended to devalue metric-based methods, nor did we intend to discuss which is more/less demanding because this could be a very subjective judgment. We are just thinking of ATM as a valuable alternative, which might also move us forward toward more human-like navigation in the future.
> >
> > **Q6: obstacle avoidance is not explained in detail, and does it make it an unfair comparison compared to other methods?**
> >
> > A6: Obstacle avoidance is achieved by simply using a distance sensor to tell the agent about its distance to its surrounding obstacles, which is standard in robotics and simulation environments. We don’t think using obstacle avoidance gives unfair comparison because the compared metric-based methods internally preserve a global metric map, which serves a similar purpose to help the agent avoid obstacles.

---

> > > ### Comment · Reviewer_bznr · 2022-11-28
> > > **Response to authors**
> > >
> > > I thank the authors for their response.
> > >
> > > **Q1: A potential weakness of this work is the lack of technical originality in the proposed method. The topological map mentioned in the paper is similar to what has already been done in previous work. I’m not sure the authors should put emphasis on the topological mapping part, both in the title, the abstract, and the introduction, as there is actually no contribution on this side. Even in the experiment part, only the short Section 5.4 is about using the topical map. It is actually not clear whether the gain on the navigation task is about the use of a topological map, or simply the fact that the underlying exploration policy covers more of the scene. The architecture of the exploration policy is also standard, as well as the imitation learning objective. The only novelty could be the additional loss term to regularise the output of the motion planner. However, if the gain from this loss term is clear (see next remark), this could be a great contribution.**
> > >
> > > Let me rephrase my concern. I understand that authors never claimed that the topological representation was a novelty but rather target the task of active data collection to build a topological map. However, as mentioned in the title, abstract and introduction, they do not claim to simply target scene exploration/coverage, but active topological mapping. However, most of the experiments conducted by the authors evaluate their active exploration method in terms of scene coverage. This is great when evaluating pure scene exploration, but according to what was introduced, more work should be put on evaluating the impact of the exploration method on the quality of the downstream topological map built from collected data. As already mentioned in my review, this is indeed done in Section 5.4, but I do think it should be a bigger part of the experimental study.
> > >
> > > **Q2: Not clear navigation gain is the use of a topological map.**
> > >
> > > This remark was related to the previous one. The gain indeed comes from the exploration method, which showcases the impact of the introduced policy. However, what would have happened if the data collected by the exploration policy was used to generate another type of scene representation? Again, the exploration policy seems efficient, but is it particularly tailored to allow better topological mapping?
> > >
> > > I think that the contribution in this work, i.e. metric-free exploration with planning in feature space, is interesting but through Q1 and Q2, I have just been questioning whether there is enough data to conclude the proposed method allows to build a better topological map (which again is the main story in the paper), or simply to maximise scene coverage (which is another important target, but quite different in my opinion).
> > >
> > > Answers from the authors  to my next questions properly addressed my concerns. I have thus increased my recommendation score to 6. I still feel the contribution is valuable, but am wondering whether the story in the paper should not be modified to put less emphasis on the active topological mapping part and more on the scene coverage maximisation. Building a better topological map could become a downstream application.

---

> > > > ### Author Response · Authors · 2022-12-06
> > > > **Further Response to the Reviewer Regarding the Experiment and Word Usage**
> > > >
> > > > We thank you for your further comments and clarification of your concerns. Your constructive comment is very well-received and important for us to improve our work. We have recently finished the experiment on ablation study regarding your question about the situation with only the action loss (L_m) and no regularisation term (L_t):
> > > >
> > > > 1. **Needs one more ablation study with the same architecture as the one proposed, but with only the action loss (L_m) and no regularisation term (L_t)?**
> > > >
> > > >   **A:** Thanks for pointing it out. First of all, what you referred to as **only the action loss (L_m) and no regularisation term (L_t)** corresponds to a variation of the ATM with supervision **ONLY** in the action space (we call it ATM_NoFeatDeepSup). To further figure out the impact of deep supervision merely in feature space, we further tested ATM_NoActDeepSup: **only the regularize  loss (L_m) and no action space deep supervision (L_t)** . The result is given in the following table:
> > > >
> > > > | Methods      | Coverage Ratio (Gibson) | Coverage Area (Gibson, $m^2$) | Coverage Ratio (MP3D) | Coverage Area (MP3D, $m^2$) |
> > > > | :-----------: | :-----------: | :-----------: | :-----------: | :-----------: |
> > > > | SPTM[1]        |0.501          | 22.268      |   0.301       |     40.121   |
> > > > | OccAnt [2]     |0.935          | 31.712      |   0.500       |     71.121   |
> > > > | ANS[3]           | **0.948**          |  32.701      |   0.521        |     73.281   |
> > > > | ATM_NoFeatDeepSup | 0.912 | 35.151 | 0.620 | 104.499 |
> > > > | ATM_NoActDeepSup | 0.900 | 33.922 | 0.600 | 102.122 |
> > > > | **ATM** | 0.918 | **35.274** | **0.642** | **109.057** |
> > > >
> > > > From this table, we can see that removing deep supervision in either feature space or action space inevitably reduces the scene coverage performance, which shows deep supervision in both feature space (task planner) and action space (motion planner) is important for efficient exploration. Moreover, it is worth noting that the performance gain via deep supervision requires no extra parameters, it thus benefits the exploration without modifying the neural network architecture.
> > > >
> > > > 2. **Your title/abstract/introduction focused on presenting active topological mapping, while the experiment section is focused on scene coverage**
> > > >
> > > > **A:**  We adopt the word "active" from the "active SLAM" in the robotics literature (e.g., this work https://arxiv.org/abs/2207.00254), “the problem of planning and controlling the motion of a robot to build the most accurate and complete model of the surrounding environment”.
> > > > In our work, since we adopt topological map, the goal of building an accurate model is reflected by the fact that we propose to use VPR to build a better graph than SPTM’s binary classification. Meanwhile, the “complete model” goal corresponds to the exploration of the unknown environment using our TAMP network.
> > > >
> > > > As the above survey paper mentioned, “Historically, active SLAM has been referred to with different terminology”, including active exploration. Considering the fact that we indeed produced a topological map via VPR after exploration, thus we adopted the word “active topological mapping”, to show the connection with previous literature as well as summarizing what our work did comprehensively.
> > > >
> > > > Given all above, we do believe our experiments on both the scene coverage (evaluation of the completeness) and the visual navigation (evaluation of the completeness and accuracy) is “enough data to conclude the proposed method allows us to build a better topological map”.
> > > >
> > > > Of course, your comment is very well received and if you indeed believe we should still update the title accordingly (and especially if this update would help you further increase your rating of our paper), we could choose wordings like ATE (Active Topological Exploration) in the final camera ready.
> > > >
> > > > [1] Nikolay Savinov et al., Semiparametric Topological Memory for Navigation, ICLR18.
> > > >
> > > > [2] Santhosh Kumar Ramakrishnan et al., Occupancy Anticipation for Efficient Exploration and Navigation. ECCV20.
> > > >
> > > > [3] Devendra Singh Chaplot et al., Learning to Explore using Active Neural SLAM. ICLR20.

---

### Official Review · Reviewer_1W7X · 2022-11-03

**Confidence:** 3
**Correctness:** 3
**Technical Novelty And Significance:** 3
**Empirical Novelty And Significance:** 3
**Recommendation:** 6

**Clarity, Quality, Novelty And Reproducibility:**

Clarity
-------
- The paper is in general well-structured. It was a pleasant read. Figures and tables are indeed very helpful to understand the method. I found the text to be sometimes a bit repetitive on some points, while skipping others which would be helpful to expand on.
- From the main text (Sec. 4) it is not clear to me how the ActionAssigner works. Why does it predict sequences, how is it trained, and its role in the mapping phase are hard to decipher.
- Some background on the employed deeply-supervised learning technique would help the reader understand the central point of joint feature extractor, TP, and MP training.
- Are the example scenes in the main text chosen from the Gibson validation set? How would they compare with MP3D test scenes (e.g. Fig. 4 and 5)? Showing some exploration and navigation examples from the more challenging MP3D test set in the main text would be informative.
- A significant number of typos is present

Quality
-------
- The technical soundness of the approach appears high. The method is described in sufficient detail and the schemes in Figures 1 to 3 are particularly heplful.
- The experimental setup is quite convincing, especially the transfer to a different domain not seen during training (MP3D)
- Testing on both coverage and navigation tasks strengthens the work
- A comprehensive quantitative evaluation and comparison of the method's overall efficiency is missing

Novelty
-------
- The work appears well-placed in the literature
- The question appears relevant and timely. Learning for TAMP is a very active field of investigation with good potential
- To my knowledge, the proposed method is novel (although I am not actively doing research in TAMP for mapping and navigation at the moment)

Reproducibility
---------
- The approach is described in sufficient detail for reproduction, with detailed material in the Appendix (only skimmed)
- Code is provided (although not checked)

Other comments
--------
- Notation-wise, in (1) and (2) it would be more correct to optimize with respect to $\pi_\theta$ rather than $\theta$
- $\theta_M$ and $\theta_T$ appear redundant and heavy in (3) and (4). Subscript $\pi_\theta$ may be enough.

**Strength And Weaknesses:**

Strengths
--------
- The integrated TAMP method is shown to learn exploration policies which can be transfered to significantly different domains, including a Gibson validation set and the more challenging MP3D scenario. Performance improvement is significant in both coverage and navigation experiments (Tab. 1 & 2). It is possible that the high-level topological representation employed by the method may favon an effective domain generalization. (As a side note, it may also be interesting to probe the performance of the TP and MP modules separately in the target MP3D domain.)

Weaknesses
---------
- Time efficiency is claimed to be one of the main advantages of the proposed method. This appears to be supported only by the number of collected images shown in Table 1 though. A comprehensive training and inference time complexity analysis and comparison with the other baselines does not seem to be reported. It would greatly strengthen the claim, ideally breaking down the individual phases, e.g., expert demos generation, pre-processing, exploration learning, and topological mapping.
- The FullHistory variant is not completely convincing in my view, since it operates with a very limited frame memory of $m=10$ as far as I understood.

**Summary Of The Paper:**

This paper presents an imitation learning-based task and motion planning (TAMP) method for active exploration and topological mapping of unknown indoor environments. Being metric-free, topological mapping allows for greater computational efficiency with respect to metric-based approaches. The method is composed of two stages: 1) Learning to actively explore from expert demonstrations; and 2) Building a topological map of the environment given the data collected at the previous step.

- Stage 1: Active exploration is achieved by joint deeply supervised learning of a panoramic scene feature extractor, a task planner, and a motion planner. In particular, the task planner (TP - a recurrent neural network) predicts/hallucinates the next image feature to be visited given a previous sequence. Then, the motion planner (MP - a multi-layer perceptron) predicts the action needed to reach/observe the hallucinated features from the current location/feature. TP is trained to minimize a regression error in feature space by minimizing an L2 loss. MP is a 3-class action classifier (turn left, turn right, move forward), trained by minimizing the cross-entropy loss. Supervision consists of demonstrations of expert explorations.

- Stage 2: Topological mapping is carried out by connecting temporally adjacent nodes retrieved from the collected exploration data, and also those matched by a visual place recognition and an efficient geometric verification technique. The corresponding agent actions are then assigned to the newly-introduced edges by a recurrent classifier, which predicts a sequence of actions.

The learned model can be employed for coverage path (and motion) planning or navigation between a starting and a target point. Experiments demonstrate the performances of the method (trained on the Gibson environment using an automated expert demonstration policy) on both the Gibson and MP3D datasets, which comprise a large number of indoor environments. Topological mapping policies learned on Gibson are shown to be able to generalize to MP3D, suggesting good generalizability of the method.

**Summary Of The Review:**

To my knowledge, the proposed TAMP learning method appears novel, timely, and technically sound. Experiments are convincing, especially since they include two relevant task and demonstrate significantly improved generalization (Gibson validation set) and domain generalization (MP3D test set) with respect to baselines.

Claims of superior time efficiency with respect to RL-based baselines appear intuitively convincing, but are not supported by an exhaustive quantitative analysis. Still, the main contribution is a significant and well-supported methodological step forward.

---

> ### Author Response · Authors · 2022-11-19
> **Response to Reviewer 1W7X**
>
> Thank you sincerely for the great comments! We provide our detailed responses as follows:
>
> **Q1: A comprehensive training and inference time complexity analysis and comparison with the other baselines are not reported. It would greatly strengthen the claim.**
>
> A1: We don’t claim time efficiency as the main contribution of our work. We discussed the time efficiency of visual place recognition (VPR) in Sec. 4 when comparing it with existing methods such as SPTM. The quantitative result is given in Table III (page 17, Appendix Sec.). For the training and inference time of each module, we will provide them in the final version due to the limited time of the rebuttal.
>
> We show the efficiency of ATM mainly from **training data size** and **experiment configuration** perspective, which is given in Table 1. Specifically, ATM just needs **one-tenth** of the dataset required by the compared RL methods and **only requires RGB images**.
>
>
> **Q2: The FullHistory variant is not convincing since it’s still using 10 frames**
>
>
> A2: We are sorry for the potentially unclear information conveyed by the name “FullyHistory”. We name it “FullHistory” because we evenly sample 10 frames **along the entire previously explored history frames**.  We changed it to “withHistory” in our updated version.
>
>
> **Q3: How ActionAssigner works? How is it trained? Its role in mapping is hard to decipher.**
>
> A3: Detailed discussion on ActionAssigner's purpose is given in Sec. 4 in the main paper, Network architecture in Table II, and Sec. D. 2 in Appendix.
>
> Five key points to better understand ActionAssigner:
>
> 1. The topological map constructed during exploration is just temporally **one-directional** connected.
>
> 2. To represent spatial adjacency, we use VPR to **add edges** for potential node pairs that were previously unconnected but spatially close.
>
> 3. For a newly added edge, we use trained ActionAssigner to assign it with the corresponding actions (which might be one action or multiple actions, which is a hyperparameter that can be tuned) that could **move the agent from one node to the other node**.
>
> 4. With ActionAssigner, the initial topological map is completed with newly added edges and the actions that are **necessary for SPTM-like visual navigation**.
>
> 5. The ActionAssigner **takes two images as input and outputs an action list (multi-label classification, 6 actions)**. It is trained on the same dataset that is used for ATM training.  If the required action number is smaller than 6, we fill the list with STOP actions.
>
> **Q4: Background information about deeply supervised learning is needed.**
>
> A4: Thanks for the suggestion. More discussions are added in the Related Work section (see updated version).
>
> **Q5: Example scenes in the main texts are from Gibson Val?**
>
> A5: Yes, they are from Gibson val, as shown in Sec. 5. We further provide more complex MP3D exploration results in Fig. I Appendix (page 14) and supplementary exploration videos.
>
> **Q6: Typo error and notation problem.**
>
> A6: Thanks for pointing it out. We have carefully revised them. $\pi_{\theta_T}$ and $\pi_{\theta_M}$ are two subnetworks in $\pi_{\theta}$, and they have different input/output as shown in Fig. 3, so we separately use two notations.

---

> > ### Comment · Reviewer_1W7X · 2022-11-28
> > **Acknowledgment of Receipt**
> >
> > I would like to sincerely thank the Authors for their clear and satifactory responses to my comments and their improvements to the manuscript. I also apologize for the delay in my acknowledgment and kindly invite the remaining Reviewers to carefully examine and consider the Authors' responses in their assessment.
> >
> > I have no further questions at the moment. I am monitoring the conversations with the other Reviewers. I currently stand by my above-threshold score and reserve the right to update it according to the feedback to the Authors' responses by the other Reviewers.

---

### Author Response · Authors · 2022-11-19
**Meta Response**

We sincerely thank all the reviewers for their thorough and insightful comments and suggestions. We thank our reviewers for the positive feedback, such as reviewer 1W7X (**novel, timely, and technically sound; convincing experiments**),  reviewer bznr (**the core idea to hallucinate the next goals directly in feature space is very interesting and promising**),  reviewer CpyF (**the work is original and the writing is clear**), and reviewer ZEmd (**self-motivated; meaningful direction; well-written and easy-to-follow**).

 Meanwhile, we found that most key concerns are about *writing clarifications*, *experimental settings*, and *literature review*. We try our best to improve our paper based on the constructive feedback of our reviewers. We believe this paper is worthwhile to be accepted at ICLR since we are the first method to *enable metric-free exploration for active topological mapping with a brand-new feature-space task and motion planning (TAMP) strategy*. We summarize our responses as follows:

1. We gave our ablation study more descriptive and accurate names.

2. We are conducting more experiments to further show the effect of deep supervision in the feature space, the time efficiency advantage of our algorithm, and to add more navigation baselines.

3. We added relevant literature reviews on visual anticipation and local planning control and tried to find the connection and synergy between them and our current work.

4. We further clarify our two-fold novelty. Firstly, feature space planning provides us with an efficient semantic/high-level encoding of the scene that’s less affected by the noise of irrelevant pixels. This technique is not well known in the robot learning community. Secondly, we take advantage of this efficient encoding to enable deep supervision in our feature space planning, which is not computationally practical in the pixel value space. This is not a common practice in the robot planning literature so far. As reviewer bznr suggested, this deep supervision can be considered a worthy contribution to the community.

5. We have corrected some typos and improved our writing that may address some minor ambiguities, such as the expert demonstration, comparison between metric methods and our method, etc.

---

### Decision · Program_Chairs · 2023-01-20

**Decision:**

Reject

**Justification For Why Not Higher Score:**

The Meta-review is self explanatory

**Justification For Why Not Lower Score:**

-

**Metareview: Summary, Strengths And Weaknesses:**

This paper had received an unusual number of 5 reviews, which were quite mixed. The reviewers appreciated the idea to hallucinate next goals directly in feature space. However, several critical issues were raised, which the authors' response could not sufficiently answer. This concerned,
- positing the work wrt to prior work and claims of novelty, as in particular the the topological map (on which the paper heavily focuses on) cannot be concsidered as a novelty; Novelty has been assessed as incremental.
- Missing or incomplete ablations.
- Missing comparisons with prior work, in particular topological maps.
- Missing references and descriptions of relevant prior work.

These issues weighted heavily into the decision, the AC recommends rejection.

The AC would like to point out that this decision was not taken on the basis of disagreements on on the definition of terms, like the meaning of the word “active”  in vision/mapping, or others.

**Summary Of Ac-Reviewer Meeting:**

No meeting was held.